# Evaluating *E. coli* genome-scale metabolic model accuracy with high-throughput mutant fitness data

David B Bernstein[1],[†] , Batu Akkas[1], Morgan N Price[2] & Adam P Arkin[1,2],*

## Abstract

The *Escherichia coli* genome-scale metabolic model (GEM) is an exemplar systems biology model for the simulation of cellular metabolism. Experimental validation of model predictions is essential to pinpoint uncertainty and ensure continued development of accurate models. Here, we quantified the accuracy of four subsequent *E. coli* GEMs using published mutant fitness data across thousands of genes and 25 different carbon sources. This evaluation demonstrated the utility of the area under a precision–recall curve relative to alternative accuracy metrics. An analysis of errors in the latest (iML1515) model identified several vitamins/cofactors that are likely available to mutants despite being absent from the experimental growth medium and highlighted isoenzyme gene-protein-reaction mapping as a key source of inaccurate predictions. A machine learning approach further identified metabolic fluxes through hydrogen ion exchange and specific central metabolism branch points as important determinants of model accuracy. This work outlines improved practices for the assessment of GEM accuracy with high-throughput mutant fitness data and highlights promising areas for future model refinement in *E. coli* and beyond.

**Keywords** flux balance analysis; genome-scale metabolic model; RB-TnSeq
**Subject Categories** Metabolism; Microbiology, Virology & Host Pathogen Interaction
**Mol Syst Biol. (2023) 19: e11566**

## Introduction

The *Escherichia coli* K-12 MG1655 genome-scale metabolic model (GEM) represents one of the most well-established compendia of knowledge on a single organism's cellular metabolism. This model maps genotype to metabolic phenotype and can be used to mechanistically simulate *E. coli* growth under various gene knockouts and/or environmental chemical perturbations. The *E. coli* GEM was one of the first GEMs to be analyzed (Varma & Palsson, 1994) and has undergone iterative curation for over 20 years (Reed *et al*, 2003; Feist *et al*, 2007; Orth *et al*, 2011; Monk *et al*, 2017). Thus, it serves

as a prominent example both for the reconstruction of new GEMs for other organisms and for benchmarking our ability to quantitatively simulate metabolism at the genome-scale (Henry *et al*, 2010; Machado *et al*, 2018; Zimmermann *et al*, 2021).

Despite success in mapping the *E. coli* genome to metabolic functions, uncertainty in GEM reconstruction and analysis still generally limits our ability to accurately simulate metabolic phenotypes (Bernstein *et al*, 2021). For example, specifications of gene-protein-reaction mappings, or the chemical composition of the environment for specific experiments, can differ from researcher to researcher or computational pipeline to pipeline (Mendoza *et al*, 2019). Furthermore, it is not always clear how to optimally simulate metabolic flux in the cell given regulatory and other nonmetabolic constraints. As we continue to reconstruct GEMs for new organisms, these issues are more prominent (Ankrah *et al*, 2021).

Critical assessment of model prediction accuracy, using experimental data, is essential for pinpointing sources of model uncertainty and ensuring continued development of accurate models. One rich source of data that can be used to validate GEMs is high-throughput mutant phenotype measurements—as measured through random barcode transposon-site sequencing (RB-TnSeq) (Wetmore *et al*, 2015; Price *et al*, 2018). This approach utilizes the power of highly parallelized genetic library screens to assay the fitness of gene knockout mutants across an array of conditions. The data that are generated can be readily simulated by GEMs and have been used recently to curate metabolic models (diCenzo *et al*, 2019; Ong *et al*, 2020), benchmark several new automated GEM reconstruction pipelines (Machado *et al*, 2018; Zimmermann *et al*, 2021) and suggest new putative gap-filled reactions (Vayena *et al*, 2022).

In this work, we provide a critical assessment of *E. coli* genome-scale metabolic models' accuracy using high-throughput mutant phenotype data measuring the fitness of *E. coli* gene knockout mutants for thousands of genes grown across environments containing 25 different primary carbon sources (Wetmore *et al*, 2015; Price *et al*, 2018). We compare the size and accuracy of the four latest *E. coli* GEMs to outline progress in the field (Reed *et al*, 2003; Feist *et al*, 2007; Orth *et al*, 2011; Monk *et al*, 2017). We then perform a detailed investigation of the errors in the latest *E. coli* GEM (iML1515). We highlight robust metrics for quantification of model accuracy, identify important adjustments to the representation of the data in the simulation of the model, highlight key

---

1 Department of Bioengineering, University of California, Berkeley, CA, USA
2 Environmental Genomics and Systems Biology Division, Lawrence Berkeley National Laboratory, Berkeley, CA, USA
*Corresponding author. Tel: +1 5104952116; E-mail: aparkin@lbl.gov
†Present address: Department of Electrical and Biomedical Engineering, University of Vermont, Burlington, VT, USA

areas for ongoing model refinement, and use a machine learning framework to suggest specific fluxes associated with incorrect model predictions.

# Results

### Progression of *E. coli* genome-scale metabolic models

We calculated the accuracy of the *E. coli* GEM by comparing model predictions to previously published experimental data (Wetmore *et al*, 2015; Price *et al*, 2018). We generated model predictions for each experiment by knocking out the specified gene and adding the specified carbon source to the simulation environment and simulating a growth/no-growth phenotype with flux balance analysis (FBA). We then quantified the accuracy of the model based on the area under a precision–recall curve (AUC) (Fig 1A and B; see Materials and Methods for additional details). The precision and recall calculations for this metric focused on true negatives (defined as experiments with low fitness and model predicted gene essentiality). This metric was chosen, as opposed to overall accuracy or the area under a receiver operating characteristic curve, because the highly imbalanced nature of the dataset (far more positives than negatives; Fig 1A inset) suggests that the correct prediction of gene essentiality is more biologically meaningful than the converse prediction of gene nonessentiality.

We began by comparing the accuracy of four versions of the *E. coli* GEM, which have been subsequently curated from 2003 to 2017 (iJR904, iAF1260, iJO1366, and iML1515) (Reed *et al*, 2003; Feist *et al*, 2007; Orth *et al*, 2011; Monk *et al*, 2017). We observed that the number of genes matched between the model and the dataset has steadily increased (Fig 1C). This indicates the increasing power of genome-scale metabolic models to capture metabolic functions. Our initial calculation of the accuracy of the models, as measured by the precision-recall AUC, showed that accuracy has steadily decreased (Fig 1D). Note that this decreasing accuracy trend was later reversed by corrections we made to the analysis approach (Fig EV2). We further demonstrated the utility of the precision–recall AUC by calculating the accuracy of each subsequent model alongside a null model where nonfunctional genes (fitness value of 0, knockout has no effect on simulated growth/no-growth) were added to the accuracy calculation (Fig EV1). The precision–recall AUC was robust to the null model relative to the alternative metrics, and thus able to capture the decreasing accuracy trend that we observed.

### Investigation of errors with the iML1515 model

We next sought to investigate the major sources of inaccuracies by closely examining the latest *E. coli* GEM (iML1515). Several key areas contributing to poor performance were apparent upon visualization and analysis of the model predictions (Fig 2, Appendix Fig S1).

First, many genes involved in vitamin and cofactor biosynthesis were leading to false-negative predictions (Fig 2A). A total of 21 different genes involved in the biosynthesis of biotin, R-pantothenate, thiamin, tetrahydrofolate and NAD$^+$ were implicated. These genes, when knocked out of the model, create a growth defect. However,

the experimental fitness of the corresponding gene knockouts was high. These predictions could be corrected by adding the vitamins/cofactors to the simulation environment. The addition of each individual vitamin/cofactor improved model accuracy, and the addition of all led to substantial improvement in accuracy (Fig 2B). This result indicates that the identified vitamins/cofactors may be available to the mutants in the RB-TnSeq experiments and suggests that the decrease in accuracy arising from these genes could arise from incorrect representation of the experimental data in the simulation environment rather than errors in the models.

Possible mechanisms for the availability of vitamins/cofactors in the experiments include cross-feeding between the diverse library of *E. coli* mutants or carry-over within individual *E. coli* mutant cells. We examined an alternative set of experimental RB-TnSeq data, collected at five and 12 generations for *E. coli* grown in a minimal glucose medium (Price *et al*, 2016). These data showed that the phenotypes for genes in the biosynthetic pathways of R-pantothenate (panB, C), thiamin (thiC-H), and NAD$^+$ (nadA-C) had weak negative fitness after five generations, but strong negative fitness after 12 generations (Appendix Table S1). This pattern supports the carry-over hypothesis and suggests that increasing the number of experimental generations could correct these false-negative predictions. Alternatively, genes in the biosynthetic pathways for biotin (bioA-D, F, H) and tetrahydrofolate (pabA, B) showed weak negative fitness at both five and 12 generations (Appendix Table S1). After 12 generations, these metabolites would be depleted by around a factor of $2^{12}$ (> 1,000x). This suggests that carry-over alone could not maintain the observed growth in these mutants. A separate study—using the Keio collection of individual gene knockout mutants across 30 carbon sources—reported that knockouts of these genes in the biotin and tetrahydrofolate pathways were not essential when assayed on solid medium (where diverse neighboring colonies could in principle cross-feed metabolites) but were essential when grown in individual liquid cultures (Tong *et al*, 2020). This suggests that biotin and tetrahydrofolate (or precursors of these metabolites) are cross-fed between *E. coli* mutants. Further in line with the carry-over and cross-feeding hypotheses, it has been demonstrated that many vitamin/cofactor precursors are stable and persist for several generations in *E. coli* and other organisms and that diverse auxotrophs' growth is supported by coculture with prototrophs (Hartl *et al*, 2017; Ryback *et al*, 2022). Considering this evidence, the cross-feeding and carry-over hypotheses should be considered when assessing the accuracy of GEM reconstruction pipelines and implementing gap-filling approaches using high-throughput mutant phenotyping data. For example, if these metabolites are present in the experiments but not added to the simulation environment, it could lead to the addition of new gap-filled biosynthetic reactions that introduce false-positive predictions in more well-controlled environments.

Next, we focused on genes that were contributing to false-positive predictions. These did not cause a growth defect when knocked out of the model but had low experimental fitness values. Three genes in the L-serine biosynthesis pathway (serA, serB, and serC) were implicated (Fig 2C). Through examination of the metabolic flux in the gene knockout models, it was observed that L-serine was being produced from glycine by a reversible reaction (GHMT2r, glycine hydroxymethyltransferase). Setting this reaction to be irreversible corrected the essentiality predictions for the three genes in the L-

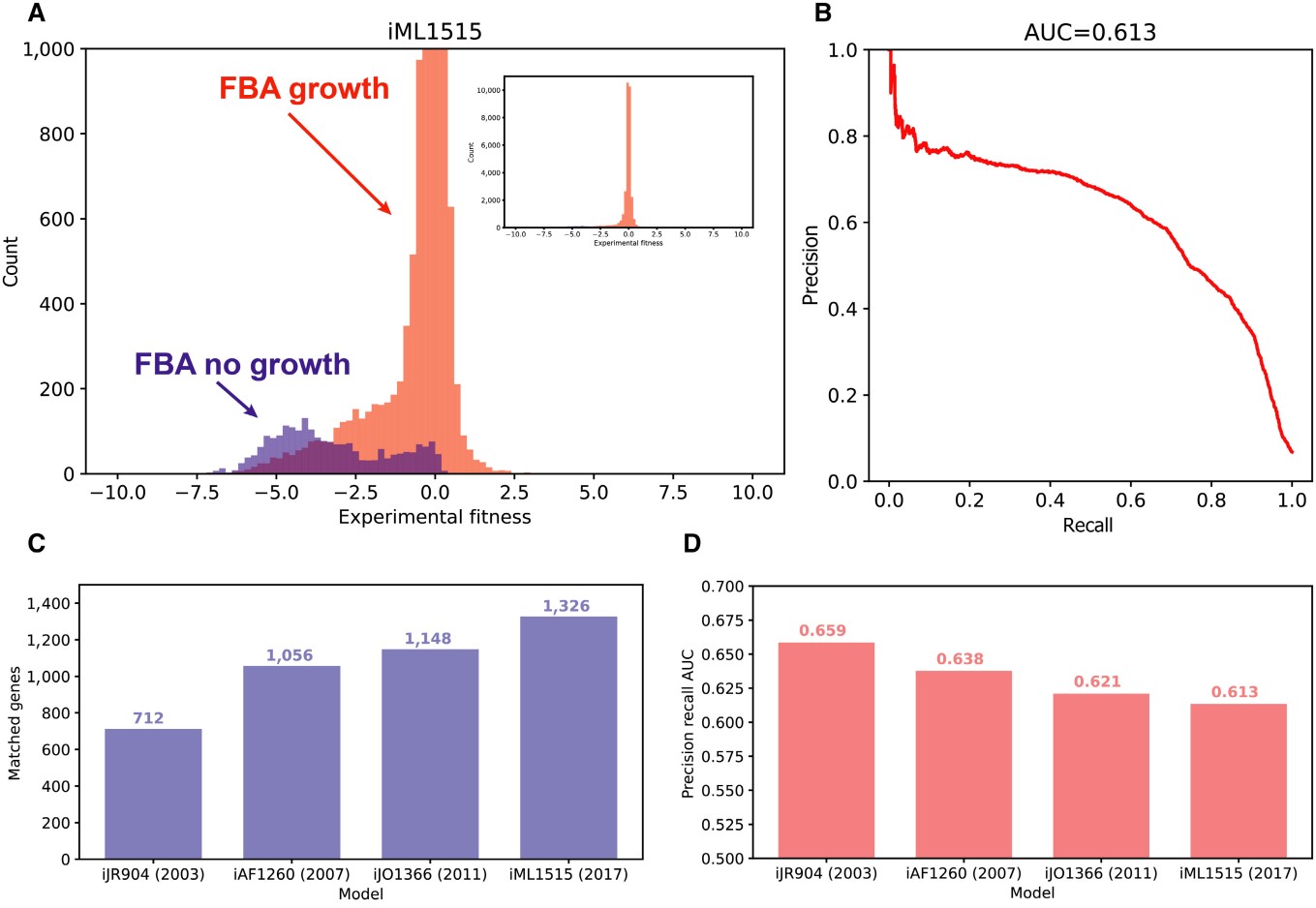

**Figure 1. Comparison of *E. coli* GEM size and accuracy for four subsequent versions.**

A  A histograms of model predictions and experimental fitness data are used to visualize the accuracy of the model. Predictions with flux balance analysis (FBA) biomass flux < 0.001 (no-growth) are included in the blue histogram, and ≥ 0.001 (growth) in the red histogram. The results for the iML1515 model are shown here. The histogram is cut off at 1,000 counts, and the inset (cutoff at 10,000 counts) shows the full histogram.

B  The area under a precision–recall curve (AUC) is used to quantify model prediction accuracy. The precision–recall curve is calculated using the fitness value as a threshold to predict model essentiality. The iML1515 curve is shown.

C  The number of genes matched between the model and the experimental dataset across subsequent *E. coli* GEMs is shown.

D  The accuracy of the models is shown across subsequent *E. coli* GEMs, as measured by area under precision-recall curve.

serine biosynthetic pathway and improved the overall accuracy of the model (Fig 2D). The reaction in question here has been proposed to run in reverse as a possible route for L-serine biosynthesis from glycine. However, the function of this reaction for this purpose is not firmly established (Price *et al*, 2020).

Another set of genes that were observed to contribute to false-positive predictions were genes involved in isoenzyme gene-protein-reaction mappings (where there is an "or" relationship in the Boolean mapping of genes to a reaction). Eight different isoenzymes mapping to 10 different reactions were among the lowest fitness genes for which the model simulated a growth phenotype (lueB, aroE, cysK, metE, ilvA, thrA, aroK, and metC; Fig 2E). Reassigning the gene-protein-reaction mapping for each of these isoenzyme/reaction pairs, such that each of the false-positive genes identified, was solely responsible for the reactions in which it is an isoenzyme, improved model performance for all but one pair (metC, CYSDS). The metC gene is mapped to two reactions as an

isoenzyme (CYSDS and CYSTL). Only the CYSTL reaction is essential in minimal carbon medium. Adjusting the isoenzyme mapping for the essential CYSTL reaction improved model accuracy but adjusting the mapping for the CYSDS had no effect on accuracy. Reassigning all isoenzyme gene-protein-reaction mappings (excluding metC, CYSDS) led to a further increase in model accuracy (Fig 2F). This correction suggests that isoenzyme representation is an important source of inaccuracy. Isoenzymes can be difficult to properly account for in metabolic models, as different enzymes may be expressed under different regulatory states (Ihmels *et al*, 2004; Jacobs *et al*, 2017). Thus, representations of isoenzymes that do not account for regulatory information can generate overly promiscuous metabolic networks leading to false-positive predictions. One example is shown in Fig 2E where dmlA can replace the function of leuB. However, dmlA expression is induced by the presence of D-malate (greater than 50-fold relative to expression in L-malate, D-glucose, or glycerol) (Stern & Hegre, 1966). Thus, dmlA would not rescue

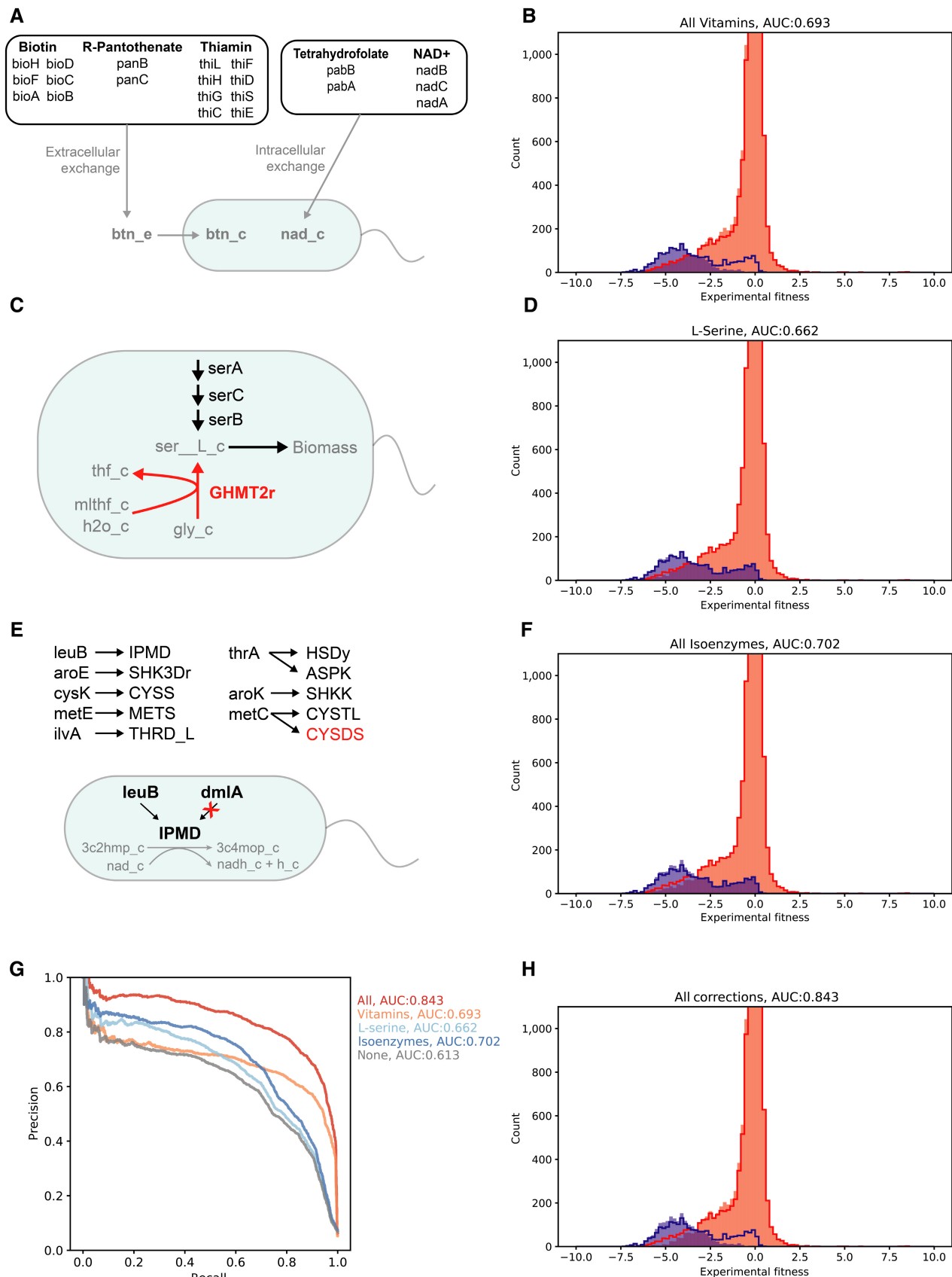

Figure 2.

**Figure 2.    Investigation of errors with the latest *E. coli* GEM (iML1515).**

A    Adding vitamins and cofactors to the model environment corrected false negatives (high fitness, model essential). Several genes in vitamin/cofactor biosynthesis pathways were among the highest average fitness with model predicted essentiality across all carbon sources. These genes are listed, grouped by their biosynthetic pathway. Vitamins/cofactors are further grouped into extracellular exchange and intracellular exchange based on whether the model contained a transporter for the associated extracellular metabolite. These vitamins/cofactors were added to the model extracellular or intracellular space through existing or newly added exchange reactions.

B    The model prediction accuracy, with vitamins/cofactors added, is displayed as the fitness histogram. The area under the precision–recall curve (AUC) is listed in the title. The original histogram, without vitamins/cofactors added, is shown as red and blue outlines. The addition of vitamins/cofactors corrected many false-negative predictions, as seen by the decrease in the component of the blue histogram with high fitness values.

C    L-serine biosynthesis gene essentiality predictions are corrected by adjusting the reversibility of the GHMT2r reaction. Three genes in the L-serine biosynthetic pathway (serA, serC, and serB) had incorrect false-positive predictions (low fitness, model nonessential). Negative flux through the GHMT2r reaction creates an alternative route for L-serine biosynthesis. Adjusting this reaction to be irreversible makes the L-serine biosynthetic genes essential and corrects the false-positive predictions.

D    The model prediction accuracy, with the GHMT2r reaction made irreversible, is displayed as the fitness histogram. The area under the precision-recall curve (AUC) is listed in the title. The original histogram, with GHMT2r reversible, is shown as red and blue outlines. The adjustment of GHMT2r reversibility corrected false-positive predictions as shown by the slight decrease in the red histogram below the red outline for low fitness values.

E    Adjustment of isoenzyme gene-protein-reaction mapping corrected false-positive predictions for several genes. Isoenzyme genes, with low fitness and model predicted nonessentiality, are listed with an arrow pointing to the reactions for which they are an isoenzyme. Gene-protein-reaction mapping was adjusted to make these isoenzymes solely responsible for their corresponding reactions. An example where the lueB gene is mapped to the IPMD reaction by removing the alternative mapping of the dmlA gene to this reaction is shown. Adjusting the gene-protein-reaction mapping improved model prediction accuracy for all isoenzymes, excluding metC to CYSDS which had no impact (shown in red).

F    The model prediction accuracy, with adjusted isoenzyme gene-protein-reaction mapping, is displayed as the fitness histogram. The area under the precision–recall curve (AUC) is listed in the title. The original histogram, with original isoenzyme gene-protein-reaction mapping, is shown as red and blue outlines. The adjustment of isoenzyme mapping corrected false-positive predictions as shown by the slight decrease in the red histogram below the red outline for low fitness values.

G    The precision–recall curve is shown for the original analysis of the model, each of the above corrections, and with all the corrections combined. The area under the precision-recall curves is listed in the legend to the right of the figure.

H    The model prediction accuracy, with all corrections combined, is displayed as the fitness histogram. The area under the precision-recall curve (AUC) is listed in the title. The original histogram, with no corrections, is shown as red and blue outlines. The corrections led to a decrease in both false-positive and false-negative predictions.

leuB mutants in many conditions. Regulation is not the only possible explanation for the incorrect isoenzyme representations. The isoenzymes could have a catalytic function with reduced efficiency that is not captured by the flux balance analysis model. Alternatively, the essential gene could have a separate essential function, such as removing a toxic substrate, that is not captured by the model. For aroE, the isoenzyme (ydiB) is known to catalyze the shikimate dehydrogenase reaction at a lower efficiency and with broader substrate and cofactor specificity; thus, its natural substrate and biological function remain unclear (Michel *et al*, 2003). For metE, the isoenzyme (metH) requires the vitamin cobalamin, which may not be present (González *et al*, 1992). For ilvA, the isoenzyme tdcB is involved in threonine degradation, rather than isoleucine synthesis, and is not expressed under the same conditions as ilvA (Egan & Phillips, 1977). Contradictorily, for aroK, the isoenzyme (aroL) has been reported to show 100-fold higher catalytic activity and is proposed as the native enzyme for the shikimate kinase reaction (DeFeyter & Pittard, 1986).

Altogether, the three corrections mentioned above, none of which is carbon source specific, substantially improved overall model prediction accuracy (Fig 2G and H, Appendix Fig S2). Note that correcting even a small number of genes here led to relatively large changes in the precision–recall AUC, as the correction for one gene generally corresponded with correcting the phenotype prediction across all 25 carbon sources. These corrections nearly eliminated false-negative predictions and substantially reduced false-positive predictions. Importantly, they point to several specific areas of GEM reconstruction where adjustments can be made to improve correspondence between model predictions and experimental data. Considering these corrections, we re-examined the trend in model accuracy for the four subsequent versions of the *E. coli* model

that we had previously observed. We recalculated the precision–recall AUC for each model while adding the vitamins/cofactors to the simulation environment (biotin, R-pantothenate, thiamin, tetrahydrofolate, and $NAD^+$ were all added through intracellular exchange) and/or excluding all isoenzymes from the data (genes with "or" statement in any associated gene-protein-reaction mapping) (Fig EV2). Notably, the addition of vitamins/cofactors improved model accuracy across all models and reversed the downward trend in model performance, highlighting the importance of this change in the simulation implementation for interpreting model accuracy results. This result is in part explained by the fact that the more recent models include more of the vitamins/cofactors (or their products) in the biomass equation and correctly represent the biosynthetic genes as conditionally essential (Appendix Table S2). The exclusion of isoenzymes from the dataset also led to an overall increase in accuracy across all models, highlighting isoenzyme representation as a general area for model improvement. This exclusion of isoenzymes did not reverse the downward trend in model performance but did give the latest iML1515 model a higher accuracy score than its predecessor. Together, both changes further improved model accuracy across all models and reversed the downward trend in model performance.

## Carbon source-specific predictions

The dataset utilized here assays 25 different carbon sources, providing insight across diverse carbon utilization pathways. We explored the carbon source-specific accuracy of the corrected iML1515 model by calculating the precision–recall AUC for each separate carbon source (Fig 3A). We observed that several carbon sources have notably lower accuracy. These patterns may indicate that our

knowledge of these specific carbon source utilization pathways is incomplete or that they have additional non-metabolic physiology that is not captured by the model, such as allosteric regulation involved in glycolysis and gluconeogenesis (Link *et al*, 2013). Additionally, we observed that carbon source-specific gene knockout model essentiality predictions were more likely to occur in genes coding for reactions that are near the specified carbon source in the metabolic network (Fig 3B). This is expected as genes in the pathway for utilization of specific carbon sources are likely to be important for growth on those substrates. It suggests that these genes are the main contributors to carbon source-specific predictions rather than global metabolic processes.

### Machine learning suggests flux profiles associated with incorrect predictions

Genome-scale metabolic modeling provides additional insight beyond a prediction of growth/no-growth. Each simulation, where a growth phenotype is predicted, simultaneously predicts the metabolic flux through every reaction in the network. We sought to use this flux information to gain deeper insight into model accuracy. We began by calculating the metabolic fluxes for each simulation using parsimonious flux balance analysis (Lewis *et al*, 2010). Visualization of the flux space for each simulation, through principal component analysis, revealed centers for each carbon source wild-type flux distribution surrounded by clouds of the gene knockout simulations grown with that carbon source (Fig 4A). The Euclidean

distance of the knockout flux vector from the wild-type flux had a slight negative correlation with the experimental fitness value (Fig 4B). This indicated a weak relationship suggesting that gene knockouts that perturb wild-type flux more greatly led to larger fitness defects.

To gain additional insight into which flux profiles are contributing most to the accuracy of the model, we used a machine learning approach. We used a gradient boosting decision tree framework, lightGBM (Ke *et al*, 2017), to classify experiments as true positives or false positives based on their simulated flux profile. Our analysis here focused on positive predictions, as negative predictions (model simulated no-growth) do not have corresponding fluxes to use as the input for the machine learning. The machine learning accuracy was assessed for repeated train/test splits with cross-validation on all experiments (carbon source and gene combinations) that were held out of the training set, and on an orthogonal test set consisting only of experiments with new carbon sources and genes that were not used in the training set (Fig 4C, Appendix Fig S3). The orthogonal test set provides a measure of the model's ability to capture metabolic processes that generalize to unseen carbon sources and genes. While this machine learning approach had weak performance, it was able to classify samples better than random, and capture general metabolic processes (Fig 4C).

Next, we utilized Shapley additive explanations, SHAP values (Lundberg & Lee, 2017; Lundberg *et al*, 2020), to quantify the importance of different fluxes in the machine learning (Fig 4D). This analysis revealed flux distributions that were associated with correct or

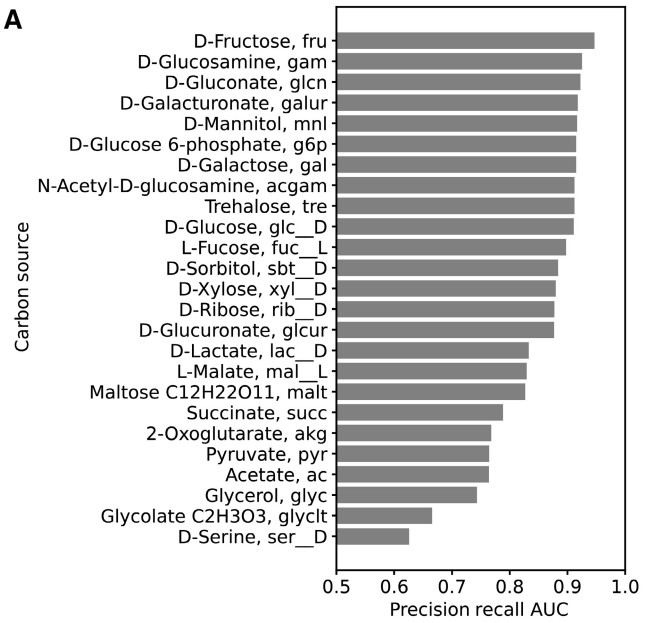

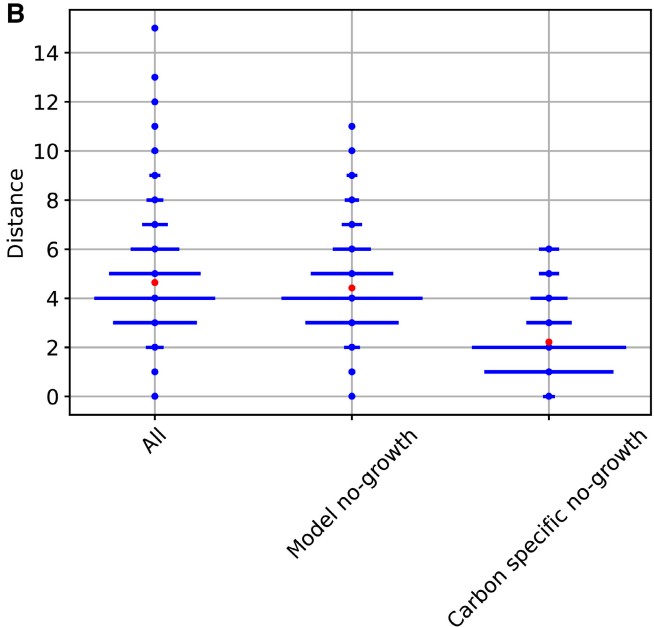

**Figure 3. Carbon source-specific metabolic model predictions.**

A  The accuracies of the model predictions (area under the precision-recall curve, with all corrections) for each specific carbon source are shown.

B  The distance along the metabolic network between the carbon source and knocked out gene is shown for different subsets of experiments. The distance is calculated as the number of reactions from carbon source to the closest reaction for which the gene is essential (see Materials and Methods for additional details). The distance is plotted as a horizontal line at each integer distance value with width proportional the fraction of experiments at that distance, and a red dot indicating the mean distance. The "all" subset of data shows the distribution of distances for all experiments involving gene knockouts that disrupt at least one reaction. The "model no-growth" subset shows the distribution of experiments with simulated no-growth. The "carbon specific no-growth" subset shows the distribution of experiments where there was simulated no-growth that was carbon source specific (no growth in 80% of carbon sources or less).

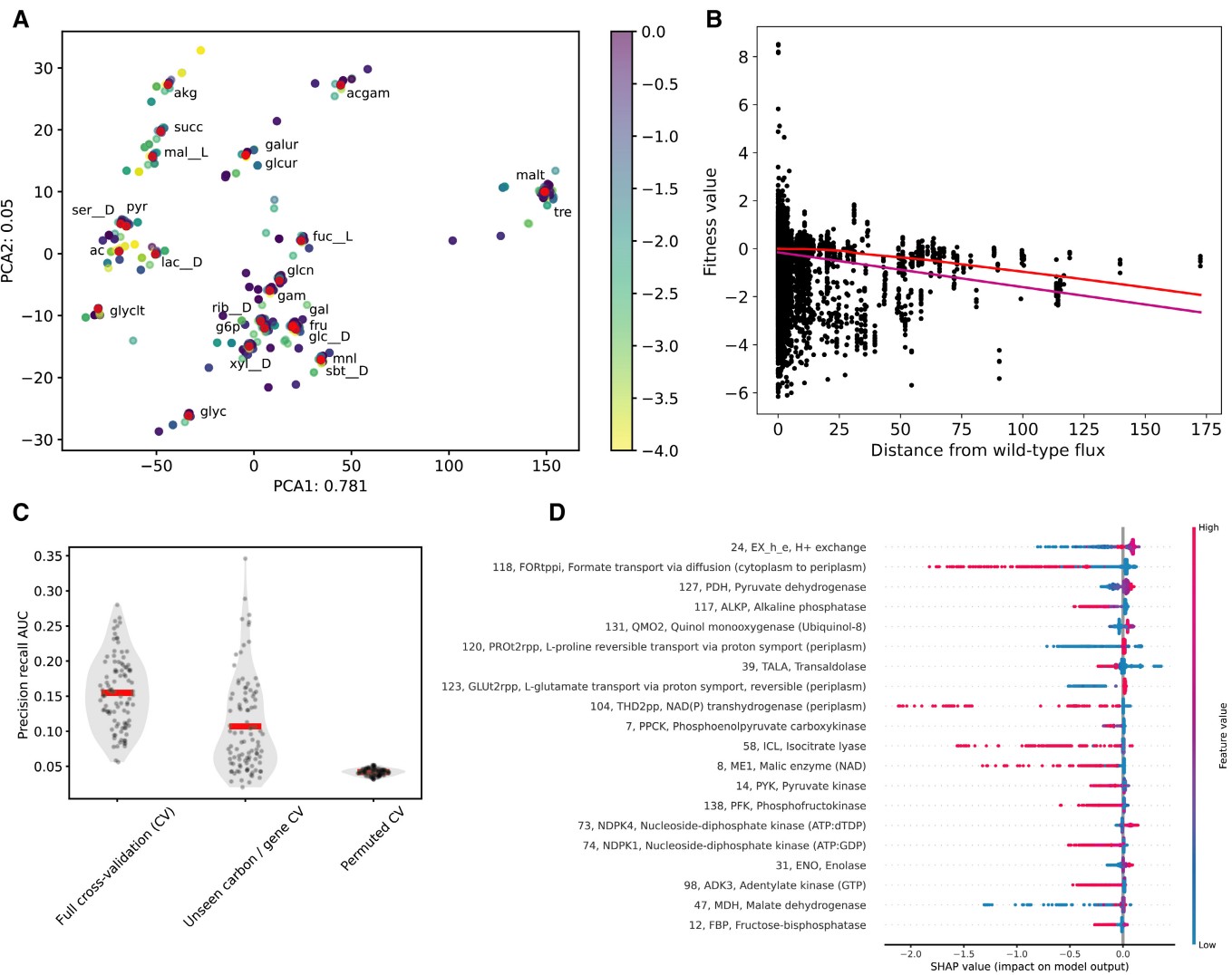

**Figure 4.  Machine learning with metabolic fluxes to investigate *E. coli* GEM (iML1515) false-positive predictions.**

A   The principal component analysis plot of experiments grouped by parsimonious flux balance analysis simulated metabolic flux is shown. Only experiments with simulated biomass flux > 0.001 are included. Wild-type carbon source fluxes (w/no gene knockouts) are shown as red points. Points are colored according to experimental fitness value. Note that points with fitness < −2 are classified as false positives throughout this machine learning analysis.

B   The plot of experimental fitness value as a function of the Euclidean distance of simulated fluxes from the wild-type fluxes on the same carbon source is shown. A slight negative correlation is seen between fitness and distance (Pearson ρ = −0.175). The purple line shows a linear fit to the data, and the red line shows a lowess fit.

C   The cross-validated accuracy of the machine learning algorithm using simulated fluxes to classify experiments as false positives (fitness values < −2) or true positives (fitness values ≥ −2) is shown. Only experiments with simulated growth (positives) are used here since the simulated fluxes are used as the input features for the machine learning algorithm. Precision–recall area under the curve is calculated for prediction of false positives. The prediction accuracy was calculated under two different cross validation schemes. In full cross-validation, these data are randomly split into train and test sets, and in the unseen carbon/gene CV, these data are split such that the train and test sets do not overlap in any gene or carbon source. See Appendix Fig S3 for additional explanation of cross-validation scheme. The permuted cross-validation shows prediction accuracy with experimental fitness values for the test set randomly permuted. Violin/scatter plots and means (red line) are shown for 100 random train/test splits.

D   The feature importance for prediction of data points as true positives calculated through SHAP is shown. Features are sorted by mean absolute SHAP value. For each feature, each sample is shown as a colored point, with the SHAP value on the x-axis and the feature value displayed through the color. When high feature value correlates with high SHAP value, this indicates that high flux through this feature is associated with correct model predictions (true positives). Inversely, high feature values correlating with low SHAP values indicates that high flux through this feature is associated with incorrect model predictions (false positives). SHAP values are averaged across 100 random train/test splits. Feature reaction index, identifier, and name are shown. Each feature corresponds to a representative flux from a flux cluster (see Materials and Methods for additional information).

incorrect predictions of the model. Several notable patterns are highlighted (Appendix Fig S4). The most important feature was the flux of hydrogen ions into or out of the cell. The machine learning

model suggested that a hydrogen ion exchange flux close to 0 was associated (through SHAP) with correct model predictions, a large positive hydrogen ion exchange (ions leaving the cell) was

associated with incorrect model predictions, and a large negative hydrogen ion flux (ions entering the cell) was associated strongly with incorrect model predictions (Appendix Fig S4A). Several of the other most informative features were also involved in hydrogen ion transfer between the periplasmic and cytoplasmic compartments of the model. The NAD(P) transhydrogenase (THD2pp) uses a flux of hydrogen ions from the periplasm to cytoplasm to reduce NADP$^+$ to NADPH using NADH. High flux through THD2pp was associated with incorrect model predictions. Two symporter reactions (PROt2rpp and GLUt2rpp) transport either L-proline or L-glutamate from the periplasm to cytoplasm along with a hydrogen ion. High negative flux through these reactions (transporting amino acids and hydrogen ions from the cytoplasm to the periplasm) was associated with incorrect model predictions. The flux through both of these reactions was also clustered (strongly covaried across simulations) with a sodium ion symporter that carried the opposite flux transporting the amino acid and a sodium ion back into the cytoplasm. Thus, the net flux of these reaction clusters is the export of hydrogen ions from cytoplasm to periplasm and import of sodium ions. To further address the hydrogen ion flux, we resimulated flux balance analysis growth predictions while fixing the hydrogen ion flux (ranging between 0 and 10) (Fig EV3). Fixing the hydrogen ion flux to a small positive value made genes in the succinate dehydrogenase complex (sdhA-D) essential on acetate and genes in the cytochrome bo complex (cyoA-D) essential on glycolate, increasing or having little effect on the model precision-recall AUC. Further constraining the hydrogen ion flux to higher values sharply decreased model accuracy by introducing false-negative predictions. Beyond the corrections identified by tuning the hydrogen ion flux, there may be more fundamental corrections to GEMs that can be implemented through a more careful representation of hydrogen ion fluxes and cross-membrane gradients.

There were also several reactions involved at branch points in central carbon metabolism that were implicated in the SHAP feature importance analysis. Increased flux through pyruvate dehydrogenase, directing pyruvate to the TCA cycle, was associated with correct model predictions (Appendix Fig S4B). Increased flux of glyceraldehyde 3-phosphate through lower glycolysis was associated with correct predictions, while negative flux through lower glycolysis was associated with incorrect predictions (Appendix Fig S4C). Alternatively, increased flux of glyceraldehyde 3-phosphate through transaldolase in the pentose phosphate pathway was associated with incorrect predictions (Appendix Fig S4D). We constrained flux through the transaldolase reaction (TALA) to 0 and found a modest increase in model accuracy (precision–recall AUC of 0.844 vs. 8.43 when unconstrained). Together, these results suggest that hydrogen ion flux, as well as several major branch points in central carbon metabolism are global determinants of model prediction accuracy.

## Discussion

In this work, we used high-throughput mutant fitness data to conduct a comprehensive analysis of the accuracy of the *E. coli* GEM. The *E. coli* GEM is a prominent example for metabolic model curation, and the dataset we utilize for validation is one of the largest consistent data sets quantifying microbial phenotypes. We demonstrated the utility of the area under a precision–recall curve as an accuracy metric and the importance of including vitamins/cofactors in the simulation environment for correct interpretation of model accuracy. Our results also pointed to reaction reversibility and isoenzyme gene-protein-reaction mapping as important areas of uncertainty in GEM reconstruction. Furthermore, a machine learning analysis identified remaining sources of model inaccuracy in hydrogen ion flux, and several key branch points in central metabolism.

It is important to note that the corrections we implement in our analysis are not necessarily the only adjustments that could correct the false model predictions we have identified. In the addition of vitamins/cofactors to the model simulation environment, it is possible that alternative precursors, rather than the vitamins/cofactors themselves, may be the metabolites being cross-fed or carried-over. For example, because the only genes implicated in the tetrahydrofolate analysis were pabA and pabB it is likely that an upstream precursor such as 4-aminobenzoate (PABA) is cross-fed rather than tetrahydrofolate. Next, while we believe the vitamin/cofactor predictions can be explained by the cross-feeding and carry-over hypotheses, other examples of false-negative predictions may alternatively be due to missing or unknown biosynthetic reactions that need to be gap-filled. For the false-positive predictions, corrected by re-assignment of reaction reversibility and isoenzyme mapping, it is possible that corrections to alternative reactions further up in the pathways of interest could correct these errors. Additionally, in our machine learning analysis, we used flux profiles from parsimonious flux balance analysis. Alternative methods for simulating metabolic flux in knockout strains have been shown to provide more accurate predictions of metabolic flux (Segrè *et al*, 2002; Shlomi *et al*, 2005; Brochado *et al*, 2012). The choice of flux simulation approach could potentially change the findings of our machine learning analysis. However, this analysis focused on identifying fluxes indicative of false-positive predictions. These fluxes enable feasible growth flux when the solution should be infeasible (reflecting a change in the shape of the solution space). Thus, this result is less likely to be impacted by these alternative approaches that change the optimization approach rather than the shape of the solution space.

Our analysis of trends in the *E. coli* GEM accuracy over time point toward the importance of the accuracy metric and simulation approach in interpreting model accuracy results. Model accuracy has been addressed in past curations of GEMs, including for *E. coli*. The original iML1515 publication assessed model accuracy using a similar gene essentiality dataset to the one used in this work. In this previous work, the authors measured the growth of the Keio collection of *E. coli* mutants across 16 different carbon sources and compared the results to iJO1366 and iML1515 model predictions (Monk *et al*, 2017). The reported overall accuracies (iML1515 accuracy: 93.4%, iJO1366 accuracy: 89.9%) are close to the overall accuracy that we calculate from our dataset when we set an experimental growth/no-growth threshold of fitness to $-2$ (iML1515 accuracy: 93.8%, iJO1366 accuracy: 92.8%). We believe that the precision–recall AUC metric we used in this work is more biologically meaningful as it emphasizes model accuracy in predicting gene essentiality. For example, the addition of a gene to the model with no metabolic function (that has no impact on the model or experiments) would yield an additional true-positive prediction for each carbon source. This nonfunctional gene would be weighted equally with a gene that is essential for growth across all carbon sources under the overall accuracy metric, while the addition of the

nonfunctional gene would have little impact on the precision–recall AUC metric. This concept is demonstrated quantitatively in our analysis of accuracy under a null model (Fig EV1).

All together, we believe that an increased focus on GEM prediction accuracy, and improved practice for reliably calculating this accuracy, will help enable GEMs to deliver on the promise of predicting phenotype from genotype. A recent community survey of metabolic modeling and microbiome researchers highlighted model trust/validation as an important focus for moving the field forward (Ankrah *et al*, 2021), and improved quantification of accuracy will help to provide this validation/trust. Beyond our work here, it will be important to establish systematic and quantitative methods for scoring the likelihood of different model corrections based on correspondence with experimental data, prior knowledge from the literature, and parsimony. Such a method could be implemented in a Bayesian framework to formalize the reconstruction/ curation of GEMs. For example, we found modest increases in model performance by constraining the model hydrogen ion flux or transaldolase reaction. Automated systematic identification of reaction constraints that improve model performance could provide additional insight and model improvements. Furthermore, curation of GEMs with experimental data and literature evidence should embrace a "deep curation" approach where different sources of evidence that converge to inform a particular model parameter are cross-validated against each other (Macklin *et al*, 2020). An exciting avenue for efficiently moving the field forward would be to engage the community in an open competition centered on critical assessment of microbial phenotype predictions, which would serve to improve integrated data collection and model accuracy quantification standards.

As we move forward, we should keep in mind that the development of models of cellular physiology that go beyond metabolism is likely necessary. While GEMs currently offer an appealing balance between predictive power and complexity, expanding models to deal with gene regulation, and other cellular processes, has the potential to further improve prediction accuracy for both metabolic and nonmetabolic phenotypes (Goldberg *et al*, 2018). Our results highlight isoenzyme gene-protein-reaction mapping as a key area for future genome-scale metabolic model improvement. There is a long history of including gene expression and/or regulation in metabolic models that could already potentially address these errors (Covert *et al*, 2004). These models range in complexity from metabolism and expression models (O'Brien *et al*, 2013) to whole cell models (Goldberg *et al*, 2018). Additionally, our machine learning results point to hydrogen ion exchange flux as a key determinant of model inaccuracy. Gradient-dependent hydrogen ion exchange is not represented in flux balance analysis, as metabolite concentrations are not represented. Modifying metabolic models to represent these gradients could be a promising route toward more accurate models of metabolic physiology. Efforts to systematically evaluate the accuracy of these more complex approaches using datasets such as the one evaluated here could providing further insight into the relationship between model complexity and accuracy. The lessons learned from the work done here related to quantifying accuracy and representing the experimental data would apply to such analyses. Furthermore, the specific areas of uncertainty identified from this work point toward areas where more complex models could provide the most return, potentially enabling the development of

more efficient models. In any case, standardized assessments of model prediction accuracy will continue to be essential for the successful application of computational modeling.

# Materials and Methods

### Computational analysis

All the methods used throughout this analysis are documented in a reproducible Python Jupyter Notebook, which is available on GitHub at github.com/dbernste/E_coli_GEM_validation. Metabolic model adjustments and simulations were conducted using COBRApy (Ebrahim *et al*, 2013). Data analysis, visualization, and machine learning were conducted using: jupyterlab (Kluyver *et al*, 2016), matplotlib (Hunter, 2007), numpy (Harris *et al*, 2020), pandas (McKinney, 2010), scipy (Virtanen *et al*, 2020), Scikit-learn (Pedregosa *et al*, 2011), lightgbm (Ke *et al*, 2017) and SHAP (Lundberg *et al*, 2020).

### Data processing

Experimental RB-TnSeq data were collected from the online fitness browser fit.genomics.lbl.gov (Price *et al*, 2018). The data from *E. coli* BW25113 were used for this analysis. The fitness values (rather than the *t* scores) were used to represent the fitness, as preliminary analyses indicated that these scores corresponded more closely with model predictions. Genes were matched to the *E. coli* GEM through their "sysName," which corresponds to their BiGG database identifiers (King *et al*, 2016). Carbon sources were matched by manually searching the BiGG database for metabolite identifiers, and other media components were similarly matched to the BiGG database. The two carbon sources sucrose and mannitol were excluded from this analysis because of known issues in the experimental preparation of their media. The fitness data for sucrose have since been removed from the fitness browser, and data for mannitol have been replaced with corrected experimental data (Price *et al*, 2022). All carbon source experiments were conducted in duplicate in the original data. The fitness scores for these duplicates were averaged for comparison with metabolic model predictions. Preliminary analysis indicated that this averaging slightly improved correspondence between fitness values and model predictions.

### Genome-scale metabolic model simulation and adjustment

Metabolic models were downloaded from the BiGG database in sbml format (King *et al*, 2016). Models were loaded and analyzed using COBRApy (Ebrahim *et al*, 2013). Models were matched to media and carbon source metabolites and exchange bounds were adjusted to add metabolites to the environment. Exchange lower bounds were set to $-1{,}000$ [mmol/(gdw × h)] to allow unlimited uptake of all noncarbon media components and $-10$ [mmol/(gdw × h)] for all carbon sources. The noncarbon media components consisted of phosphate, sulfate, ammonium, oxygen, hydrogen ions, water, carbon dioxide, and metal ions (BiGG IDs: pi; co2; fe3; h; mn2; fe2; zn2; mg2; ca2; ni2; cu2; sel; cobalt2; h2o; mobd; so4; nh4; k; na1; cl; o2; tungs; slnt). Models were adjusted to account for differences between the experimental BW25113 *E. coli* strain and the model

MG1655 strain by removing several genes and their corresponding reactions that are not present in the BW25113 strain (Grenier *et al*, 2014). Models were further adjusted to remove nonconditionally essential genes from the analysis (genes that were essential when all possible exchanges have lower bound set to −1,000). Many of these nonconditionally essential genes were already removed from the fitness data as they do not have reliable phenotype measurements in RB-TnSeq experiments due to low representation in the initial library. Therefore, it is possible that the remaining nonconditionally essential genes, matched between model and dataset, could be false-negative predictions. We chose to remove all these genes from our analysis for consistency. Model gene knockouts were simulated using cobrapy function model.genes.id.knock_out, which uses the Boolean gene-protein-reaction mapping to remove reactions from the model based on gene knockouts. Model simulations were conducted with model.slim_optimize, which provided substantial speed improvements when only recording the biomass flux, or with parsimonious FBA through cobra.flux_analysis.pfba to record all simulated fluxes (Lewis *et al*, 2010). Parsimonious FBA was run with default parameters such that the solution found minimizes the sum of total flux while maintaining the optimal growth flux. A biomass flux of 0.001 [gdw/(gdw × h)] was used throughout this analysis as a growth/no-growth cutoff.

Several corrections were made to the analysis of the iML1515 model to improve accuracy. Vitamins/cofactors were added to the simulation environment. In total, five vitamins/cofactors were added first individually and then all together. These metabolites were either added to the simulation environment either directly to the cytoplasmic compartment (biotin, R-pantothenate, thiamin) or to the extracellular compartment (tetrahydrofolate, and $NAD^+$) if the model already had the corresponding transport and exchange reactions. The lower bound of these exchange reactions were set to −1,000 to allow unlimited uptake of these metabolites. A correction was made by changing the reversibility of a reaction in the L-serine biosynthesis pathway (GHMT2r, glycine hydroxymethyltransferase). This reaction was made irreversible by setting the lower bound equal to 0. Isoenzyme gene-protein-reaction mappings were corrected for several isoenzymes (lueB, aroE, cysK, metE, ilvA, thrA, aroK, and metC). These genes were identified to be leading to false-positive predictions. Each gene-protein-reaction mapping was changed such that each of these genes was solely responsible for any reaction where they had previously been mapped through an or relationship with another gene.

## Model accuracy calculation

Model accuracy was calculated using the area under a precision–recall curve. The model simulated FBA biomass flux data were binarized to a growth/no-growth phenotype based on a cutoff of 0.001 (no-growth < 0.001, growth ≥ 0.001). A sliding threshold on the fitness value was then used to generate a precision–recall curve. The positive class was set to simulated essentiality (no-growth phenotype). The area under this precision–recall curve was used to quantify model accuracy. Precision–recall curves were calculated using sklearn.metrics.pre_rec and area under the curve was calculated using sklearn.metrics.auc.

Several alternative accuracy metrics were calculated for comparisons with the area under a precision–recall curve metric. The area under the receiver operating characteristic curve was calculated using sklearn.metrics.roc_auc_score, balanced accuracy was calculated as the arithmetic mean of the true-positive rate and true-negative rate, and the overall accuracy was calculated as the sum of true positives and true negatives divided by the total number of samples.

A null model was used to improve the interpretation of model accuracy results. This null model assumed that all additional genes in a subsequent model were nonfunctional (fitness = 0, no effect on simulated growth/no-growth). This analysis was performed by adding null data points to each model until the total number of experiments (number of genes multiplied by number of carbon sources) matched that of the subsequent model. The accuracy was then calculated, with each metric, while including these null data points.

## Carbon source distance analysis

The distance between carbon sources and genes was calculated using the genome-scale metabolic model. The model was converted to a bipartite graph with metabolites and reactions as nodes. An edge was placed between any metabolite and reaction where that metabolite was a reactant or product for that reaction (nonzero stoichiometry), and an adjacency matrix was constructed for the metabolic network. From this adjacency matrix, high degree "hub" nodes were removed. Hub nodes were defined as any node with more than 50 connections and consisted of central metabolites such as hydrogen and water, cofactors such as ATP and $NAD^+$, and the *E. coli* Biomass reactions. Pyruvate was also a hub node but was left in the network due to its role as a carbon source in this analysis. This hub-less network was then used to calculate the carbon source gene distance. The distance was calculated as the number of reactions traversed from the extracellular carbon source to the nearest reaction that was essentially encoded by the gene (a gene knockout removed the reaction). For example, distance of 0 corresponded to an essential reaction directly involving the extracellular carbon source, while distance of 1 corresponded to an essential reaction involving any metabolite connected to a reaction of distance 0. Distances on the metabolic network were calculated using the scipy.csgraph.shortest_path method implementation of Dijkstra's algorithm.

## Machine learning and feature importance

Machine learning (ML) was conducted to classify simulations with biomass flux into false positives (experimental fitness < −2) or true positives (experimental fitness ≥ −2). Note that we also ran the analysis with a fitness threshold of −1 and found our main results to be robust to this choice; overall, there was a Spearman's rank correlation of 0.93 between feature importance calculations with the −2 or −1 threshold. A lightgbm classifier was used for machine learning. Flux vectors, quantifying the simulated flux across all reactions in the metabolic model (2,714 total fluxes), were used as the input for the ML algorithm. Fluxes with variance across samples less than $10^{-7}$ were discarded form the analysis (leaving 579 total fluxes). Fluxes were clustered into groups of fluxes with covariation greater than 0.99 across samples (leaving 172 flux clusters). One representative flux from each cluster was used for the ML input. ML accuracy was assessed by 100 repeated train/test splits. For each train/test, split a

random subset of samples from 80% of the carbon sources and 80% of the genes was selected as the training set. Cross-validated accuracy was assessed on the full set of test samples as well as a smaller set of samples with no overlapping carbon sources or genes in the training set. Model performance was explored for varying number of leaves in the ML model, an important tuning parameter for overfitting. In the final model, five leaves were used, which balanced full cross-validation with new carbon/gene cross-validation. Other parameters were set to the lightgbm classifier default values (boosting_type = "gbdt", num_leaves = 5, max_depth = −1, learning_rate = 0.1, n_estimators = 100, subsample_for_bin = 200,000, objective = None, class_weight = None, min_split_gain = 0.0, min_child_weight = 0.001, min_child_samples = 20, subsample = 1.0, subsample_freq = 0, colsample_bytree = 1.0, reg_alpha = 0.0, reg_lambda = 0.0, random_state = None, n_jobs = None, importance_type = "gain"). Feature importance was calculated using Shapley additive explanations (SHAP) through shap.-TreeExplainer (Lundberg *et al*, 2020). Shapley additive explanations values shown in the text are for the classification of samples as true positives and averaged across train/test splits.

## Data availability

(i) Code: Available on GitHub (https://github.com/dbernste/E_coli_GEM_validation); (ii) Fitness data: Available on GitHub (https://github.com/dbernste/E_coli_GEM_validation/tree/main/Fitness_Data/E_coli_BW25113) and through the fitness browser (https://fit.genomics.lbl.gov/cgi-bin/org.cgi?orgId = Keio) (Price *et al*, 2018); (iii) Metabolic models: Available on GitHub (https://github.com/dbernste/E_coli_GEM_validation/tree/main/Models) and through the BiGG database (http://bigg.ucsd.edu/models/iJR904, http://bigg.ucsd.edu/models/iAF1260, http://bigg.ucsd.edu/models/iJO1366, http://bigg.ucsd.edu/models/iML1515) (King *et al*, 2016).

**Expanded View** for this article is available online.

## Acknowledgements

We would like to acknowledge helpful discussion and feedback from all members of the Arkin lab. The contributions of DBB and APA were supported by the Consortium for Monitoring, Technology, and Verification under U.S. Department of Energy National Nuclear Security Administration, award number DE-NA0003920. The contributions of APA and MNP by Ecosystems and Networks Integrated with Genes and Molecular Assemblies (ENIGMA; http://enigma.lbl.gov), a Science Focus Area program at Lawrence Berkeley National Laboratory, is based upon work supported by the U.S. Department of Energy, Office of Science, Office of Biological & Environmental Research under contract number DE-AC02-05CH11231.

## Author contributions

**David B Bernstein:** Conceptualization; data curation; software; formal analysis; supervision; validation; visualization; methodology; writing – original draft; writing – review and editing. **Batu Akkas:** Data curation; software; formal analysis; visualization; writing – review and editing. **Morgan N Price:** Data curation; supervision; validation; methodology; writing – review and editing. **Adam P Arkin:** Conceptualization; supervision; funding acquisition; methodology; project administration; writing – review and editing.

## Disclosure and competing interests statement

The authors declare that they have no conflict of interest.

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
