## [Review Process File · Molecular Systems Biology]

Evaluating *E. coli* genome-scale metabolic model accuracy with high-throughput mutant fitness data

David Bernstein, Batu Akkas, Morgan Price, and Adam Arkin

DOI: [10.15252/msb.202311566](https://doi.org/10.15252/msb.202311566)

Corresponding author(s): Adam Arkin (aparkin@lbl.gov)

Review Timeline:

Submission Date:	1st Feb 23
Editorial Decision:	13th Mar 23
Revision Received:	18th Aug 23
Editorial Decision:	22nd Sep 23
Revision Received:	23rd Sep 23
Accepted:	5th Oct 23

Editor: Maria Polychronidou

Transaction Report:

13th Mar 2023

Manuscript Number: MSB-2023-11566, Critical assessment of E. coli genome-scale metabolic model with high-throughput mutant fitness data

Dear Prof Arkin,

Thank you again for submitting your work to Molecular Systems Biology. We have now heard back from the three reviewers who agreed to evaluate your study. As you will see below, the reviewers acknowledge that the presented findings seem relevant. They do however raise a series of concerns, which we would ask you to address in a major revision.

Without repeating all the points listed below, some of the most essential issues include those raised by reviewer #2 (points 1-3), who mentions that additional analyses need to be performed in order to provide more concrete recommendations on improving genome-scale model performance. During our cross-commenting process, in which the reviewers are given the chance to (anonymously) comment on each other's reports, reviewer #3 pointed out that they share the concerns raised by reviewer #2 and think that they need to be convincingly addressed.

All issues raised by the referees would need to be satisfactorily addressed. As you may already know, our editorial policy allows in principle a single round of major revision. It is therefore essential to provide responses to the reviewers' comments that are as complete as possible. Please let me know in case you would like to discuss in further detail any of the issues raised, I would be happy to schedule a call.

On a more editorial level, we would ask you to address the following points:

- Please provide a .doc version of the manuscript text (including legends for the main figures) and individual production quality figure files for the main Figures (one file per figure).
- We have replaced Supplementary Information by the Expanded View (EV format). In this case, all additional figures and Tables can be included in a PDF called Appendix. Appendix figures and Tables should be labeled and called out as: "Appendix Figure S1, Appendix Figure S2... Appendix Table S1..." etc. Each legend should be below the corresponding Figure/Table in the Appendix. Please include a Table of Contents in the beginning of the Appendix. For detailed instructions regarding expanded view please refer to our Author Guidelines: .
- Please provide a "standfirst text" summarizing the study in one or two sentences (approximately 250 characters), three to four "bullet points" highlighting the main findings and a "synopsis image" (550px width and max 400px height, jpeg format) to highlight the paper on our homepage.
- All Materials and Methods need to be described in the main text. We would encourage you to use 'Structured Methods', our new Materials and Methods format. According to this format, the Material and Methods section should include a Reagents and Tools Table (listing key reagents, experimental models, software and relevant equipment and including their sources and relevant identifiers) followed by a Methods and Protocols section in which we encourage the authors to describe their methods using a step-by-step protocol format with bullet points, to facilitate the adoption of the methodologies across labs. More information on how to adhere to this format as well as downloadable templates (.doc or .xls) for the Reagents and Tools Table can be found in our author guidelines: . An example of a Method paper with Structured Methods can be found here:
- Please include a "Disclosure & Competing Interests Statement".
- Please include a "Data availability" section describing how the data, code, models etc. have been made available. This section needs to be formatted according to the example below:
The datasets and computer code produced in this study are available in the following databases:
 - Chip-Seq data: Gene Expression Omnibus GSE46748 (<https://www.ncbi.nlm.nih.gov/geo/query/acc.cgi?acc=GSE46748>)
 - Modeling computer scripts: GitHub (<https://github.com/SysBioChalmers/GECKO/releases/tag/v1.0>)
 - [data type]: [full name of the resource] [accession number/identifier] ([doi or URL or identifiers.org/DATABASE:ACCESSION])
- For data quantification: please specify the name of the statistical test used to generate error bars and P values, the number (n) of independent experiments (specify technical or biological replicates) underlying each data point and the test used to calculate p-values in each figure legend. The figure legends should contain a basic description of n, P and the test applied. Graphs must include a description of the bars and the error bars (s.d., s.e.m.).
- Molecular Systems Biology supports formal data citations in the Reference list, to cite previously published datasets. In addition to citing the original papers that reported the data, we encourage you to also cite the relevant datasets directly in the Reference

list. In the text, references to datasets are included as "Data ref: Smith et al, 2001" or "Data ref: NCBI Sequence Read Archive PRJNA342805, 2017". In the Reference list, data citations are very similar to normal literature references but must be labeled with "[DATASET]" at the end of the reference. For detailed instructions please refer to our Author Guidelines .

- The Reference list should be formatted according to the Molecular Systems Biology reference style. The links to the published papers (e.g. <https://doi.org/10.1038/s41592-019-0686-2>) should be removed from the Reference list.

- When you resubmit your manuscript, please download our CHECKLIST (<https://bit.ly/EMBOPressAuthorChecklist>) and include the completed form in your submission.

Please note that the Author Checklist will be published alongside the paper as part of the transparent process (<https://www.embopress.org/page/journal/17444292/authorguide#transparentprocess>).

If you feel you can satisfactorily deal with these points and those listed by the referees, you may wish to submit a revised version of your manuscript. Please attach a covering letter giving details of the way in which you have handled each of the points raised by the referees. A revised manuscript will be once again subject to review and you probably understand that we can give you no guarantee at this stage that the eventual outcome will be favorable.

Kind regards,

Maria

Maria Polychronidou, PhD
Senior Editor
Molecular Systems Biology

We realize that it is difficult to revise to a specific deadline. In the interest of protecting the conceptual advance provided by the work, we recommend a revision within 3 months (11th Jun 2023). Please discuss the revision progress ahead of this time with the editor if you require more time to complete the revisions. Use the link below to submit your revision:

IMPORTANT: When you send your revision, we will require the following items:

1. the manuscript text in LaTeX, RTF or MS Word format
2. a letter with a detailed description of the changes made in response to the referees. Please specify clearly the exact places in the text (pages and paragraphs) where each change has been made in response to each specific comment given
3. three to four 'bullet points' highlighting the main findings of your study
4. a short 'blurb' text summarizing in two sentences the study (max. 250 characters)
5. a 'thumbnail image' (550px width and max 400px height, Illustrator, PowerPoint or jpeg format), which can be used as 'visual title' for the synopsis section of your paper.
6. Please include an author contributions statement after the Acknowledgements section (see <https://www.embopress.org/page/journal/17444292/authorguide>)
7. Please complete the CHECKLIST available at (<https://bit.ly/EMBOPressAuthorChecklist>).

Please note that the Author Checklist will be published alongside the paper as part of the transparent process (<https://www.embopress.org/page/journal/17444292/authorguide#transparentprocess>).

See also figure legend guidelines: <https://www.embopress.org/page/journal/17444292/authorguide#figureformat>

9. Please note that corresponding authors are required to supply an ORCID ID for their name upon submission of a revised manuscript (EMBO Press signed a joint statement to encourage ORCID adoption).

(<https://www.embopress.org/page/journal/17444292/authorguide#editorialprocess>)

Currently, our records indicate that there is no ORCID associated with your account.

Please click the link below to provide an ORCID:

Link Not Available

The system will prompt you to fill in your funding and payment information. This will allow Wiley to send you a quote for the article processing charge (APC) in case of acceptance. This quote takes into account any reduction or fee waivers that you may

be eligible for. Authors do not need to pay any fees before their manuscript is accepted and transferred to the publisher.

EMBO Press participates in many Publish and Read agreements that allow authors to publish Open Access with reduced/no publication charges. Check your eligibility: <https://authorservices.wiley.com/author-resources/Journal-Authors/open-access/affiliation-policies-payments/index.html>

*** PLEASE NOTE *** As part of the EMBO Press transparent editorial process initiative (see our Editorial at <https://dx.doi.org/10.1038/msb.2010.72>), Molecular Systems Biology publishes online a Review Process File with each accepted manuscripts. This file will be published in conjunction with your paper and will include the anonymous referee reports, your point-by-point response and all pertinent correspondence relating to the manuscript. If you do NOT want this File to be published, please inform the editorial office at msb@embo.org within 14 days upon receipt of the present letter.

Reviewer #1:

Overall, a very solid and highly useful dataset. I am not convinced by the interpretation of the data-simulation duo and strongly recommend revisions as the current phrasings and shortcomings in interpretation/discussion may (unnecessarily) negatively bias the reader and future studies.

1. "the cross-feeding and carry-over hypotheses should be considered 156 when assessing the accuracy of GEM reconstruction" -> this is a key result of the study! Why is this hidden in the text and not highlighted in the Abstract/Title? As it stands the title/abstract is negatively toned. All models are wrong and some are useful - so there is no point in emphasizing shortcomings of the previous models. I am sure that there is much scope to improve the here-proposed model as well. For example, sources of errors include unannotated genes, incomplete many-to-one or one-to-many gene-function mappings etc. So, this is likely more about incomplete genetic/biochemical knowledge rather than models being inaccurate.
2. Mutants are not necessarily expected to exhibit flux optimal phenotypes. There are other simulation methods to address this (MOMA, MIMBL, ROOM etc.) and should be included / at the least mentioned as a discussion point.
3. Reaction reversibility and isoenzyme curations: is there literature evidence for the proposed changes?
4. "representation of carbon source utilization pathways is more accurate for glycolytic substrates than for other alternative pathways." This is not necessarily correct / only explanation. It could easily be that GPR mappings are more complex for central pathways which would be natural for high-flux/critical pathways. Also, it should be noted that there are several allosteric regulatory interactions in these pathways - plenty of literature on this including from Sauer and Ralser labs etc..
5. Abstract: "gold standard for the simulation of cellular 15 metabolism": not true - depends on the cell that researchers want to simulate.

Reviewer #2:

The authors present a systematic study of gene essentiality predictions with E. coli genome scale metabolic network model (GEM). Their methods are solid and interesting. I strongly agree with the idea that precision/recall is the better way to assess predictions than accuracy. The paper is readable, although I think the language could be made more clear for the general audience.

Studies of this kind may be useful as gene essentiality predictions are commonly used for benchmarking new reconstructions of metabolic networks based on flux balance analysis (FBA). The ability of a model to accurately predict essential genes by FBA indicates that biomass and energy production pathways are well annotated and properly represented. However, we all know that we cannot expect perfect predictive performance, especially for two reasons: (i) metabolism is heavily regulated, which is not visible to FBA, and (ii) metabolite concentrations (e.g., that of a toxic metabolite accumulated when an enzyme is perturbed) cannot be predicted by FBA. As an example for the former, if an essential reaction is associated with two isozymes which are tested for essentiality, without knowing which one is expressed and whether regulation can compensate for the perturbation by increasing the expression of the unperturbed isozyme, there is no way to predict essentiality of either gene. Thus, benchmarking models with gene essentiality must be done with care. One way studies like this can help is to educate researchers on proper use and interpretation of gene essentiality predictions. Another way is by annotating new pathways or correcting misannotations in a model when correcting false predictions. Unfortunately, we did not see significant contributions in this study to these or other aspects of gene essentiality analysis for the following major reasons:

1. Most of the modifications that are presented as model corrections are actually not permanent fixes on the model but are conditional changes implemented to fit the predictions to the data used. For example, the enzyme set of some reactions with multiple isozymes was reduced to one essential enzyme. We do not see how this can be useful for any future work. The removed isozymes may be unexpressed and unused under the experimental conditions tested here, but we cannot, and should not, take them out of the model just based on this data since they may be essential under other conditions depending on regulation, or they may be playing a role in the production of a useful metabolite without affecting fitness (the observed variable in this study) appreciably. Similarly, media corrections and even reaction reversibility changes are not permanent fixes, but are good approximations for a particular set of conditions. The latter (reaction reversibility) could be a permanent fix, but this requires thermodynamic analysis that takes into account expected in vivo concentrations and activity coefficients, because reactions can be conditionally reversed when products accumulate.

2. The systematic analysis of the carry over of trace metabolites and exchange of metabolites between mutants is important. However, these mechanisms are obvious, expected, and observed all around. The analysis should be published somewhere as it is carefully conducted, but whether significant enough for this journal is a question.

3. The machine learning (ML) analysis to find sources of false predictions is very interesting. However, the patterns found just show some indicators of predictive failure, like proton exchange. What do these indicators point to as sources of false predictions? How can we improve the models by fixing these sources? Without answering such questions, ML results are interesting but not significant.

Minor comments:

1. The authors use an arbitrary threshold on predicted Biomass production flux to call essential genes. This is an adequate method, but not necessarily the only way. Did the authors consider other methods such as minimization of metabolic adjustment (MOMA)? MOMA can be used to quantify the response to a perturbation as the divergence from a reference Biomass producing flux distribution (obtained by pFBA).

2. Page 5, Line 137-139: "This data showed that the phenotypes for genes in the biosynthetic pathways of R138 pantothenate (panB, C), thiamin (thiC-H), and NAD+ (nadA-C) had weak negative fitness after 5 139 generations, but fitness dropped off to be strongly negative after 12 generations". Why are there no figures for this analysis?

3. Analysis related to Figures C->D and E->F. Very few genes are affected by the modifications such as changing a reaction's reversibility or removing a few isozymes. How can these have such large effects on AUC while only a small percent of genes are affected? This needs to be discussed.

4. Text related to Figure 4E. It is not clear how isozymes are adjusted. I figured this by looking at the GPRs of reactions but the text and figure are not helpful in understanding what was done.

5. Full cross validation and new cross validation during ML are not clearly explained. Actually those parts of the text are quite confusing. Figure S3A is clear, but this is not visible enough. In particular, it sounds from text that 64% (training)+36%("full" CV)+4%("new" CV) are all from the same 100%, but they total 104%.

6. Page 16, Line 467: Exchange rates cannot be set at -1000 by default. I believe this is a typo. I would expect them to be set at 0 for carbon sources not present in the media (those in the media are said to be set at -10).

7. Page 17, Line 512: "Machine learning (ML) was conducted to classify simulations with biomass flux into false positives (experimental fitness < -2) or true positives (experimental fitness >= -2)". Why use a sharp threshold like that? I would think there is not much of a difference between -1.99 and -2.01. Why not use a conservative low threshold to call unfit, and a conservative high for fit, and exclude those in between as no call?

Reviewer #3:

Summary

This paper presents an in-depth analysis of the performance of the recently published genome-scale metabolic network reconstruction/knowledgebase of *E. coli*, iML1515. Performance is assessed in terms of how well a flux balance analysis model based on iML1515 is able to predict bacterial fitness (growth) subject to gene knockouts and a variety of carbon sources. Modifications are proposed to the flux balance analysis model that improve prediction accuracy, such as rendering certain pathways irreversible to ensure that growth is infeasible under those conditions. Lastly, a gradient boosting decision tree model is used to predict the fitness of a strain based on its simulated flux profile.

The overall finding is dramatic: "We observed that model size is increasing while prediction accuracy is decreasing." When I read this I felt like I could almost hear it in Adam's voice (eyebrow firmly arched)! This statement is something people have noticed or suspected for many years, and so the effort to identify and quantify it should be commended. I was also impressed by the focus on hydrogen ion exchange, and the ability to connect specific flux values with correct or incorrect predictions. I do have some suggestions of course, but by and large I find this paper to be of interest to the broader metabolic modeling community.

Positive remarks

The paper demonstrates a good approach to rectify inaccuracies in metabolic models to better fit experimental data. Highlighting the non-obvious issues with accuracy metrics for metabolic models was interesting to read, and could be elaborated further upon.

Major revisions

Figure 1D, it would be convenient to see how your improved iML1515 P/R AUC compares to the original iML1515. In addition, a major point made in the paper is that accuracy decreases as model size increases, is this still true after your model corrections are made across all four GEMs?

Fig. 4C For the experiments predicting fitness from simulated flux profiles with the gradient boosted decision tree, a precision-recall AUC ranging around 0.05-0.15 might be too weak for the predictions to be of much use. I would try to find another way to compare the flux profiles - maybe extending the PCA to include the false positives, using SVD, or something third. Before predicting fitness, a simple clustering of the different "failure modes" could be useful. Also, adding a sentence to flesh out the point of "permuted CV" would be helpful.

Line 58, it seemed like this sentence glossed over the fact that adding regulatory and other non-metabolic constraints now has a decades-long history, and in fact has led to many improved predictions.

Line 71, there are many studies in which knockout strain phenotypes, measured in different environments, have been compared to FBA predictions - it's a standard of the field. Palsson, Covert, Segré, and others have all done work in exactly this space, so it's probably worth a compare and contrast to establish and explain the novelty here (maybe novel environments or strains that haven't been previously compared).

Line 89, I really liked this part - too often FBA comparisons focus on all of the predictions in a unified metric, whereas in reality the negative predictions and experimental observations matter more. Well put! That said, it might be helpful to *also* include the total accuracy, just in case there's anything else in there (false positives etc, although I do see you cover some of these later in the paper). In such a case you might compare the model predictions to random shuffling of the measured phenotypes to see if the FBA predictions are statistically significant - this would be better than a simple comparison of % correct predictions.

Line 99, there must be caveats to this because of a trade-off - larger models can be more inclusive and have a larger breadth of predictions, while smaller models can't predict as much but are more accurate. This also brings up a larger concern: why weren't any of the earlier models compared to iML1515? Do the findings you make have any impact on the earlier models? The way that the paper is set up, I was expecting to see more and more failures propagate over time, but that seems to be completely missing.

Line 126, I found this part unconvincing, because correcting for the environment is not really a modeling error. You might think of it as a "simulation error". The reactions were all correct, the S matrix was correct, but at first you didn't get all the environmental inputs correct. Some clarification on what "essential gene" exactly means in this context is needed.

Line 141, increasing the number of experimental generations is a great idea. In one of the whole-cell modeling papers (Karr 2012), one of the key findings was that some knockouts were only lethal after several generations. This to me seemed like it could explain why sometimes experimentally it's possible to get colonies of E. coli mutants that never grow up to a culture. The knockout isn't lethal at first. So, very interesting to see it here. That said, I wouldn't say that increasing the number of generations is correcting "model error" - that's probably still "simulation error."

Line 169, "This correction..." I would say that checking the reversibility of certain reactions to improve model predictions is already a pretty well-known technique.

Line 184, This was done pretty extensively in Covert Nature 2004 - adding regulatory information to the Reed E. coli model, then comparing 13,750 experimentally measured phenotypes to rFBA predictions (110 conditions x 125 strains, biologic data). Sure enough, adding regulation helped a lot, especially with isoenzymes as you suggest!

Line 334, I liked this part with transaldolase - might be nice to follow up on it - I'm sure knockouts exist, and probably even published work on knockouts - it's trivial to knock it out in FBA. What happens? It seems as though it would be productive and possibly even valuable to know of genes that could be knocked out to "calm the system down" or make it more predictable and less noisy. Just a thought.

Line 464 Since flux profiles simulated by flux balance analysis are a large part of the paper, it would be helpful if it was made clear in a succinct manner what the constraints and objectives look like, and how the constraints are modified by e.g. knockouts and media changes. For example, these are a few model considerations I was not able to easily find in the Materials and Methods section (although sometimes I could infer it from other sections):

*For a gene knockout, what exactly happens to the reaction network? Does it remove all reactions where relevant enzyme complexes include the protein produced by that gene?

*What weighting does pFBA use for the total flux minimization relative to the objective function? Does changing the weighting change the flux profile significantly?

*Other than the carbon source, what does the media contain?

Just meant to get you started - I would consider looking at Methods sections in similar papers.

Reviewer Replies:

Note: All line numbers correspond to manuscript_file_marked_up.docx

Reviewer #1:

Overall, a very solid and highly useful dataset. I am not convinced by the interpretation of the data-simulation duo and strongly recommend revisions as the current phrasings and shortcomings in interpretation/discussion may (unnecessarily) negatively bias the reader and future studies.

R1.1 1. "the cross-feeding and carry-over hypotheses should be considered 156 when assessing the accuracy of GEM reconstruction" -> this is a key result of the study! Why is this hidden in the text and not highlighted in the Abstract/Title? As it stands the title/abstract is negatively toned. All models are wrong and some are useful - so there is no point in emphasizing shortcomings of the previous models. I am sure that there is much scope to improve the here-proposed model as well. For example, sources of errors include unannotated genes, incomplete many-to-one or one-to-many gene-function mappings etc. So, this is likely more about incomplete genetic/biochemical knowledge rather than models being inaccurate.

We greatly appreciate this comment from the reviewer and agree with the sentiment. We have made significant changes to our manuscript, to refocus on emphasizing improved practices for utilizing high-throughput fitness data to quantify model accuracy rather than focusing on model quality. Indeed, our results do not necessarily demonstrate a degradation in model quality, rather they highlight improved practices for assessing model accuracy. This is made clear by a new analysis where we have re-evaluated the accuracy of the four *E. coli* GEMs while addressing the vitamin/cofactor and isoenzyme issues that we identified (Expanded View Figure EV2, Line 911). Additionally, we have added another analysis that further highlights the utility of our chosen accuracy metric, another key result outlining best practices for the analysis of genome-scale metabolic model accuracy with high-throughput mutant fitness data (Expanded View Figure EV1, Line 899). We have also updated the title to remove the negative tone from the word "critical". We have also updated the abstract to focus on improved practice for evaluating model accuracy, including further highlighting the vitamin/cofactor result (Lines 22-25). Furthermore, we updated the text throughout the introduction, results, and discussion to reflect the improved focus of the paper (Lines 38-50, 113-122, 240-256, 491-495).

R1.2 2. Mutants are not necessarily expected to exhibit flux optimal phenotypes. There are other simulation methods to address this (MOMA, MIMBL, ROOM etc.) and should be included / at the least mentioned as a discussion point.

We agree with the reviewer, that other simulation methods can more accurately predict metabolic flux for gene knockouts. However, to our knowledge these approaches work by identifying an alternative point in the solution space, rather than changing the shape of the solution space. Thus, our main accuracy results, which depend only on growth/no-growth predictions, would not be affected by choosing one of these alternative simulation approaches. It is true that the results of the machine learning

section, which utilize metabolic flux profiles, could be changed. We now mention these approaches and address this in a revised discussion (Lines 472-480).

R1.3 3. Reaction reversibility and isoenzyme curations: is there literature evidence for the proposed changes?

We do provide general support from the literature for our hypothesis that isoenzymes are expressed under different conditions and thus would not rescue each other. We also showed one example with support from the literature (leuB). In this example, leuB and dmlA both enable the essential IPMD reaction, but dmlA is only expressed under certain conditions and thus cannot always rescue leuB. We have now added additional literature supported explanations related to several of the other genes (aroE, metE, and ilvA), and contradictory evidence for aroK (Lines 219-231).

R1.4 4. "representation of carbon source utilization pathways is more accurate for glycolytic substrates than for other alternative pathways." This is not necessarily correct / only explanation. It could easily be that GPR mappings are more complex for central pathways which would be natural for high-flux/critical pathways. Also, it should be noted that there are several allosteric regulatory interactions in these pathways - plenty of literature on this including from Sauer and Ralser labs etc..

We agree with the reviewer that this may not be the correct/only explanation for the observed results. We have re-worded the results to avoid making declarative statements here. We have also included reference to research from the Sauer lab demonstrating allosteric regulation in glycolysis/gluconeogenesis. (Lines 305-314).

R1.5 5. Abstract: "gold standard for the simulation of cellular 15 metabolism": not true - depends on the cell that researchers want to simulate.

We have changed the text in the abstract to remove the term "gold standard" (Line 14). Likewise, we have changed this text in the introduction (Line 60) and discussion (Line 447).

Reviewer #2:

The authors present a systematic study of gene essentiality predictions with E. coli genome scale metabolic network model (GEM). Their methods are solid and interesting. I strongly agree with the idea that precision/recall is the better way to assess predictions than accuracy. The paper is readable, although I think the language could be made more clear for the general audience.

Studies of this kind may be useful as gene essentiality predictions are commonly used for benchmarking new reconstructions of metabolic networks based on flux balance analysis (FBA). The ability of a model to accurately predict essential genes by FBA indicates that biomass and energy production pathways are well annotated and properly represented. However, we all know that we cannot expect perfect predictive performance, especially for two reasons: (i) metabolism is heavily regulated, which is not visible to FBA, and (ii) metabolite concentrations (e.g., that of a toxic metabolite accumulated when an enzyme is perturbed) cannot be predicted by FBA. As an

example for the former, if an essential reaction is associated with two isozymes which are tested for essentiality, without knowing which one is expressed and whether regulation can compensate for the perturbation by increasing the expression of the unperturbed isozyme, there is no way to predict essentiality of either gene. Thus, benchmarking models with gene essentiality must be done with care. One way studies like this can help is to educate researchers on proper use and interpretation of gene essentiality predictions. Another way is by annotating new pathways or correcting misannotations in a model when correcting false predictions. Unfortunately, we did not see significant contributions in this study to these or other aspects of gene essentiality analysis for the following major reasons:

R2.1 1. Most of the modifications that are presented as model corrections are actually not permanent fixes on the model but are conditional changes implemented to fit the predictions to the data used. For example, the enzyme set of some reactions with multiple isozymes was reduced to one essential enzyme. We do not see how this can be useful for any future work. The removed isozymes may be unexpressed and unused under the experimental conditions tested here, but we cannot, and should not, take them out of the model just based on this data since they may be essential under other conditions depending on regulation, or they may be playing a role in the production of a useful metabolite without affecting fitness (the observed variable in this study) appreciably. Similarly, media corrections and even reaction reversibility changes are not permanent fixes, but are good approximations for a particular set of conditions. The latter (reaction reversibility) could be a permanent fix, but this requires thermodynamic analysis that takes into account expected in vivo concentrations and activity coefficients, because reactions can be conditionally reversed when products accumulate.

We appreciate this comment from the reviewer, and it has helped us rework how we present and discuss the results of our work. We agree with the reviewer that changes such as updating the isoenzyme gene-protein-reaction mapping are not necessarily permanent changes. Rather, they point towards areas where the data should be analyzed with care, and avenues for potential model/simulation framework improvement. We have revised the manuscript with an increased focus on emphasizing how our work “helps to educate researcher on proper use and interpretation of gene essentiality predictions”, as suggested by the reviewer above. Updates along these lines include changing the title to remove the negative tone, updating the abstract to focus results on lessons for data analysis improved practice, as well as changes throughout the results and discussion (Lines 17-34, 38-50, 113-122, 240-256, 491-495). In addition to these changes, we have added a new analysis that further demonstrates the utility of the precision recall accuracy metric relative to alternative metrics, a key result demonstrating improved practices in data analysis (Expanded View Figure EV1, Line 899). As well as a new analysis that demonstrates the importance of our data analysis corrections for the interpretation of model accuracy trends (Expanded View Figure EV2, Line 911). Our new manuscript provides more clear guidance exemplifying improved practice for the evaluation of genome-scale metabolic model accuracy with high-throughput fitness data, rather than suggesting permanent changes to the models.

R2.2 2. The systematic analysis of the carry over of trace metabolites and exchange of

metabolites between mutants is important. However, these mechanisms are obvious, expected, and observed all around. The analysis should be published somewhere as it is carefully conducted, but whether significant enough for this journal is a question. We appreciate the reviewer's comment and complement of the careful conduct of our analysis. We agree that the carry-over hypothesis is an important result from our work. We have further emphasized this result in our abstract and synopsis (Lines 22-24, 38-50). Additionally, we have added a new analysis that demonstrates how the trend in model accuracy we observed is dependent on this correction (Results Lines 240-256, Expanded View Figure EV2, Line 911). Also mentioned in response to R2.1.

Regarding the significance of our findings, we believe that our work outlines important improved practices for the assessment of genome-scale metabolic models with high-throughput fitness data. This data is currently being used to evaluate the performance of genome-scale metabolic model reconstruction pipelines and to suggest new reactions for gap-filling models (Introduction Lines 79-82). Our work demonstrates important improved practices for these applications. We now show that the interpretation of trends in model performance can depend on these corrections (Expanded View Figure EV2, Line 911) and discuss important implications for gap-filling vitamin/cofactor biosynthesis pathways with this data (lines 180-185). Thus, we believe it is important for this work to receive high visibility as it will help other researchers carefully implement their analysis of these valuable datasets. Our work does not necessarily present a new advance in model development, but it does help move the field towards more well validated and trustworthy models which has been identified as a major hurdle facing the application of genome-scale metabolic models (Lines 515-518).

R2.3 3. The machine learning (ML) analysis to find sources of false predictions is very interesting. However, the patterns found just show some indicators of predictive failure, like proton exchange. What do these indicators point to as sources of false predictions? How can we improve the models by fixing these sources? Without answering such questions, ML results are interesting but not significant.

We agree with the reviewer's assessment that the machine learning results point to indicators of model predictive failures, rather than specific fixes that can be made to the models. We did implement one analysis to attempt to improve model accuracy based on these results, where we fixed the hydrogen ion exchange flux (Supplemental Figure S5 Line 956). We also tried fixing the transaldolase (TALA) reaction flux to 0 to improve model accuracy (Lines 404-406, also see response to R3.11). Both changes led to modest improvements, and we believe further improvements to model predictive accuracy could be made by more carefully considering the representation of hydrogen ion fluxes in metabolic models. Our publication highlights this area for ongoing effort, we have attempted to make this clearer in our discussion (Lines 522-525).

Minor comments:

R2m.1 1. The authors use an arbitrary threshold on predicted Biomass production flux to call essential genes. This is an adequate method, but not necessarily the only way. Did the authors consider other methods such as minimization of metabolic adjustment

(MOMA)? MOMA can be used to quantify the response to a perturbation as the divergence from a reference Biomass producing flux distribution (obtained by pFBA). We did look at the distance of the knockout flux from the wild-type flux and found a slight negative correlation with fitness (Figure 4B). We have also added a new discussion of MOMA and related methods to our manuscript (also in response to R1.2) (Lines 472-480).

R2m.2 2. Page 5, Line 137-139: "This data showed that the phenotypes for genes in the biosynthetic pathways of R138

pantothenate (panB, C), thiamin (thiC-H), and NAD⁺ (nadA-C) had weak negative fitness after 5

139 generations, but fitness dropped off to be strongly negative after 12 generations".

Why are there no figures for this analysis?

The original data is from Price *et al* 2016. We have now added a new table that concisely provides this data (Supplemental Table S1, Line 975).

R2m.3 3. Analysis related to Figures C->D and E->F. Very few genes are affected by the modifications such as changing a reaction's reversibility or removing a few isozymes. How can these have such large effects on AUC while only a small percent of genes are affected? This needs to be discussed.

The relatively large effect here is because these gene phenotypes are carbon source non-specific. Correcting one gene generally corrects the phenotype prediction for all 25 carbon sources. Thus, one gene correction corresponds to 25 data point corrections. We now explain this further in the results (Line 234-236). The AUC is also further improved by the fact that these predictions tend to be the "most wrong" prior to correction. The false positives have very low fitness values, and the false negatives have very high fitness values, which is captured by this metric.

R2m.4 4. Text related to Figure 4E. It is not clear how isozymes are adjusted. I figured this by looking at the GPRs of reactions but the text and figure are not helpful in understanding what was done.

We have clarified the text to improve our description of how the isoenzymes were adjusted (Lines 202-205).

R2m.5 5. Full cross validation and new cross validation during ML are not clearly explained. Actually those parts of the text are quite confusing. Figure S3A is clear, but this is not visible enough. In particular, it sounds from text that 64% (training)+36%("full" CV)+4%("new" CV) are all from the same 100%, but they total 104%.

We have modified the text in the caption of Figure 4 and added a pointer to the supplemental Figure S3 to clarify the cross-validation scheme (Lines 426-433). We have also clarified the explanation of the cross-validation scheme in supplemental Figure S3 A (Line 936-942).

R2m.6 6. Page 16, Line 467: Exchange rates cannot be set at -1000 by default. I believe this is a typo. I would expect them to be set at 0 for carbon sources not present in the media (those in the media are said to be set at -10).

The exchange bounds were set to -1000 to allow unlimited uptake of all non-carbon media components. We have now clarified this in the text. (Lines 603-608).

R2m.7 7. Page 17, Line 512: "Machine learning (ML) was conducted to classify simulations with biomass flux into false positives (experimental fitness < -2) or true positives (experimental fitness ≥ -2).". Why use a sharp threshold like that? I would think there is not much of a difference between -1.99 and -2.01. Why not use a conservative low threshold to call unfit, and a conservative high for fit, and exclude those in between as no call?

The threshold of -2 was chosen for convenience in the machine learning analysis. This threshold is typically representative of strong fitness defects; however, an alternative value could have been chosen. We have now re-run our analysis with a threshold of -1, capturing weaker fitness defects. This change in fitness threshold leads to only small changes in the feature importance ranks (Spearman correlation $\rho = 0.93$ between -2 and -1 analyses) and our major results and discussion are independent of this change in threshold value. We now report this in our manuscript methods section (Line 682-685).

Reviewer #3:

Summary

This paper presents an in-depth analysis of the performance of the recently published genome-scale metabolic network reconstruction/knowledgebase of *E. coli*, iML1515. Performance is assessed in terms of how well a flux balance analysis model based on iML1515 is able to predict bacterial fitness (growth) subject to gene knockouts and a variety of carbon sources. Modifications are proposed to the flux balance analysis model that improve prediction accuracy, such as rendering certain pathways irreversible to ensure that growth is infeasible under those conditions. Lastly, a gradient boosting decision tree model is used to predict the fitness of a strain based on its simulated flux profile.

The overall finding is dramatic: "We observed that model size is increasing while prediction accuracy is decreasing." When I read this I felt like I could almost hear it in Adam's voice (eyebrow firmly arched)! This statement is something people have noticed or suspected for many years, and so the effort to identify and quantify it should be commended. I was also impressed by the focus on hydrogen ion exchange, and the ability to connect specific flux values with correct or incorrect predictions. I do have some suggestions of course, but by and large I find this paper to be of interest to the broader metabolic modeling community.

Positive remarks

The paper demonstrates a good approach to rectify inaccuracies in metabolic models to better fit experimental data. Highlighting the non-obvious issues with accuracy metrics for metabolic models was interesting to read, and could be elaborated further upon.

Major revisions

R3.1 Figure 1D, it would be convenient to see how your improved iML1515 P/R AUC compares to the original iML1515. In addition, a major point made in the paper is that accuracy decreases as model size increases, is this still true after your model corrections are made across all four GEMs?

This was an excellent suggestion. We have run a new analysis measuring the precision-recall AUC for each of the 4 models while accounting for the major corrections we identified (Lines 240-256). We found that the downward trend in model performance is reversed when accounting for the vitamin/cofactor correction (Expanded View Figure EV2, Line 911). Accordingly, we have updated the presentation of our results to focus on highlighting improved practices for model accuracy evaluation with high-throughput fitness data, rather than focusing on model performance trends. In addition to this new analysis and figure we updated the text throughout in the abstract, synopsis, results, and discussion (Lines 17-34, 38-50, 113-122, 240-256, 491-495).

R3.2 Fig. 4C For the experiments predicting fitness from simulated flux profiles with the gradient boosted decision tree, a precision-recall AUC ranging around 0.05-0.15 might be too weak for the predictions to be of much use. I would try to find another way to compare the flux profiles - maybe extending the PCA to include the false positives, using SVD, or something third. Before predicting fitness, a simple clustering of the different "failure modes" could be useful. Also, adding a sentence to flesh out the point of "permuted CV" would be helpful.

We have updated the PCA plot in Figure 4 to utilize a smaller color range to better display the important fitness values between 0 and -4. This facilitates the interpretation of false positives on the plot (any point with fitness <-2). We also added a note to the figure caption pointing this threshold out to readers (Lines 416-417). We tried an alternative PCA plot that had separate colors for false vs. true positives and found that this did not facilitate the interpretation of the data beyond the current updated PCA plot.

We also tried looking at hierarchically clustered heatmaps to better understand the failure modes but found the data difficult to interpret this way due to the large number of features and samples. We believe the machine learning results provide better insights. We appreciate the reviewer's suggestions and feel more confident in our approach having explored this other avenue.

We have modified the text in the caption of Figure 4 and added a pointer to the supplemental Figure S3 to clarify the cross-validation scheme (Lines 422-435). We have also clarified the explanation of the cross-validation scheme in Supplemental Figure S3 A (Line 936-942).

R3.3 Line 58, it seemed like this sentence glossed over the fact that adding regulatory and other non-metabolic constraints now has a decades-long history, and in fact has led to many improved predictions.

This is a good point. We believe this should be addressed in the discussion in the context of suggested areas for model improvement. We have now added text and

references where we discuss areas for model improvement and point out that adding regulation (to fix isoenzyme representations) could potentially be addressed by these existing methods (Lines 546-551).

R3.4 Line 71, there are many studies in which knockout strain phenotypes, measured in different environments, have been compared to FBA predictions - it's a standard of the field. Palsson, Covert, Segré, and others have all done work in exactly this space, so it's probably worth a compare and contrast to establish and explain the novelty here (maybe novel environments or strains that haven't been previously compared). We believe that the novelty in our work is in our analysis approach and our focus on improved practices for model accuracy quantification. We have re-focused our manuscript to highlight these areas. Including: demonstrating the importance of the chosen accuracy metric for the interpretation of model accuracy, including vitamins/cofactors in the simulation environment and highlighting areas of model inaccuracies (isoenzyme gene-protein-reaction mapping, and hydrogen ion exchange and several central carbon metabolism pathways identified through machine learning). We do directly address the novelty of our work relative to previous quantification of the *E. coli* GEM accuracy with similar data in the discussion of our chosen accuracy metric (Line 495-511).

R3.5 Line 89, I really liked this part - too often FBA comparisons focus on all of the predictions in a unified metric, whereas in reality the negative predictions and experimental observations matter more. Well put! That said, it might be helpful to *also* include the total accuracy, just in case there's anything else in there (false positives etc, although I do see you cover some of these later in the paper). In such a case you might compare the model predictions to random shuffling of the measured phenotypes to see if the FBA predictions are statistically significant - this would be better than a simple comparison of % correct predictions.

Thanks for the positive feedback and good suggestion. We have now included an additional analysis that more closely compares different accuracy metrics (Expanded View Figure EV1, Line 899). This analysis compares accuracies as calculated by precision-recall AUC, ROC AUC, balanced accuracy, and overall accuracy. In addition, it includes a null model where non-functional genes are added to the metabolic network. We believe that this new analysis helps to demonstrate the utility of the precision-recall AUC metric and makes this a more central result of our paper.

R3.6 Line 99, there must be caveats to this because of a trade-off - larger models can be more inclusive and have a larger breadth of predictions, while smaller models can't predict as much but are more accurate. This also brings up a larger concern: why weren't any of the earlier models compared to iML1515? Do the findings you make have any impact on the earlier models? The way that the paper is set up, I was expecting to see more and more failures propagate over time, but that seems to be completely missing.

This is a good point and seems to be related to the reviewers first comment (R3.1) as well as the previous comment (R3.5). We believe that this has been addressed through

the two new analyses we mention in those comments (Expanded View Figure EV1 Line 899, and EV2 Line 911).

R3.7 Line 126, I found this part unconvincing, because correcting for the environment is not really a modeling error. You might think of it as a "simulation error". The reactions were all correct, the S matrix was correct, but at first you didn't get all the environmental inputs correct. Some clarification on what "essential gene" exactly means in this context is needed.

We agree with the reviewer's characterization of these errors as "simulation error" rather than "model error". We have re-written the text to emphasize this distinction, referring to these corrections as improved approaches for data interpretation, rather than model corrections (Lines 155-157, Figure 2 caption title Line 259-260). Nonetheless, we believe this is an important correction to point out for proper interpretation of this data through models.

R3.8 Line 141, increasing the number of experimental generations is a great idea. In one of the whole-cell modeling papers (Karr 2012), one of the key findings was that some knockouts were only lethal after several generations. This to me seemed like it could explain why sometimes experimentally it's possible to get colonies of E. coli mutants that never grow up to a culture. The knockout isn't lethal at first. So, very interesting to see it here. That said, I wouldn't say that increasing the number of generations is correcting "model error" - that's probably still "simulation error."

Again, we agree with the reviewer's interpretation. In line with the previous comment (R3.8) We have updated our manuscript throughout to refer to these corrections as improved interpretation of data or corrections to the analysis, rather than "model corrections" (Lines 139, 259-260, 295, 461, 626).

R3.9 Line 169, "This correction..." I would say that checking the reversibility of certain reactions to improve model predictions is already a pretty well-known technique.

We agree and have removed the sentences from the results suggesting that this result alone points to reaction reversibility as a key variable (Lines 193-194). We do believe that this specific reaction was important to highlight.

R3.10 Line 184, This was done pretty extensively in Covert Nature 2004 - adding regulatory information to the Reed E. coli model, then comparing 13,750 experimentally measured phenotypes to rFBA predictions (110 conditions x 125 strains, biologic data). Sure enough, adding regulation helped a lot, especially with isoenzymes as you suggest!

We have added a discussion of previous methods that incorporate regulation alongside our discussion of the isoenzyme related errors, also in response to R3.3 (Lines 546-551).

R3.11 Line 334, I liked this part with transaldolase - might be nice to follow up on it - I'm sure knockouts exist, and probably even published work on knockouts - it's trivial to knock it out in FBA. What happens? It seems as though it would be productive and

possibly even valuable to know of genes that could be knocked out to "calm the system down" or make it more predictable and less noisy. Just a thought.

This is a good idea, and nice suggestion by the reviewer. We tried knocking out transaldolase as a "correction" and found a very modest increase in model performance Precision Recall AUC 0.844 improved from 0.843 (Line 405-406). It is possible that knocking out different reactions could "calm down" the system more and further improve model performance. We have added this to our discussion and suggest that more systematic identification of flux constraints is a promising approach to provide additional insight (Lines 522-525).

R3.12 Line 464 Since flux profiles simulated by flux balance analysis are a large part of the paper, it would be helpful if it was made clear in a succinct manner what the constraints and objectives look like, and how the constraints are modified by e.g. knockouts and media changes. For example, these are a few model considerations I was not able to easily find in the Materials and Methods section (although sometimes I could infer it from other sections):

*For a gene knockout, what exactly happens to the reaction network? Does it remove all reactions where relevant enzyme complexes include the protein produced by that gene?

*What weighting does pFBA use for the total flux minimization relative to the objective function? Does changing the weighting change the flux profile significantly?

*Other than the carbon source, what does the media contain?

Just meant to get you started - I would consider looking at Methods sections in similar papers.

We appreciate this comment and the suggestions to clarify our methods. We have updated our methods to provide additional information. We list the non-carbon media components in the methods (Line 605-608). We point to the gene knockout function we used throughout the manuscript and provide a brief description (Lines 617-619). We have added a brief description of pFBA (Lines 622-624).

In lieu of additional changes, we believe that our methods point to appropriate references with specific details contained therein. Additionally, all the analysis done in this work is available unambiguously and open source through our github page, although we appreciate that it is not as efficient to parse the methods directly from the code.

22nd Sep 2023

Manuscript Number: MSB-2023-11566R

Title: Evaluating E. coli genome-scale metabolic model accuracy with high-throughput mutant fitness data

Dear Prof. Arkin,

Thank you for sending us your revised manuscript. We have now heard back from the two reviewers who were asked to evaluate your revised study. As you will see below, the reviewers are satisfied with the performed revisions and are supportive of publication. Reviewer #2 only recommends moving Appendix Figure S5 to the EV Figures. Before we can formally accept the manuscript for publication, we would also ask you to address some editorial issues listed below.

- Our data editors and I have made some minor formatting changes. Please make all requested text changes using the attached file and *keeping the "track changes" mode* so that we can easily access the edits made.

- Please include a callout to Fig. 1C.

- Please include page numbers in the Appendix Table of Contents.

Please resubmit your revised manuscript online ****within one month**** and ideally as soon as possible. If we do not receive the revised manuscript within this time period, the file might be closed and any subsequent resubmission would be treated as a new manuscript. Please use the Manuscript Number (above) in all correspondence.

Click on the link below to submit your revised paper.

Kind regards,

Maria

Maria Polychronidou, PhD
Senior Editor
Molecular Systems Biology

If you do choose to resubmit, please click on the link below to submit the revision online before 22nd Oct 2023.

IMPORTANT:

Please note that corresponding authors are required to supply an ORCID ID for their name upon submission of a revised manuscript (EMBO Press signed a joint statement to encourage ORCID adoption).

(<https://www.embopress.org/page/journal/17444292/authorguide#editorialprocess>)

Currently, our records indicate that the ORCID for your account is 0000-0002-4999-2931.

Link Not Available

The system will prompt you to fill in your funding and payment information. This will allow Wiley to send you a quote for the article processing charge (APC) in case of acceptance. This quote takes into account any reduction or fee waivers that you may be eligible for. Authors do not need to pay any fees before their manuscript is accepted and transferred to the publisher.

*** PLEASE NOTE *** As part of the EMBO Press transparent editorial process initiative (see our Editorial at <https://dx.doi.org/10.1038/msb.2010.72> , Molecular Systems Biology will publish online a Review Process File to accompany accepted manuscripts. When preparing your letter of response, please be aware that in the event of acceptance, your cover letter/point-by-point document will be included as part of this File, which will be available to the scientific community. More information about this initiative is available in our Instructions to Authors. If you have any questions about this initiative, please contact the editorial office (msb@embo.org).

Reviewer #2:

With the clarifications in text, the addition of Figures EV1 and EV2, and the good points made by authors in the responses to all reviewer comments, I am now convinced about the significance of this study. I am confident to say this paper will be a good reference for validation of GEMs based on gene essentiality analysis. I still think some points and sections could be made more clear but that is a style issue. My only suggestion is to make Fig S5 more visible by making it Fig EV3.

Reviewer #3:

Our comments have been addressed well enough I would say. Thanks, and congrats on the exciting work!

All editorial and formatting issues were resolved by the authors.

5th Oct 2023

Manuscript number: MSB-2023-11566RR

Title: Evaluating E. coli genome-scale metabolic model accuracy with high-throughput mutant fitness data

Dear Prof. Arkin,

Thank you again for sending us your revised manuscript. We are now satisfied with the modifications made and I am pleased to inform you that your paper has been accepted for publication.

*** PLEASE NOTE *** As part of the EMBO Publications transparent editorial process initiative (see our Editorial at <https://dx.doi.org/10.103/msb.2010.72>), Molecular Systems Biology publishes online a Review Process File with each accepted manuscripts. This file will be published in conjunction with your paper and will include the anonymous referee reports, your point- by-point response and all pertinent correspondence relating to the manuscript. If you do NOT want this File to be published, please inform the editorial office at msb@embo.org within 14 days upon receipt of the present letter.

LICENSE AND PAYMENT:

All articles published in Molecular Systems Biology are fully open access: immediately and freely available to read, download and share.

Molecular Systems Biology charges an article processing charge (APC) to cover the publication costs. You, as the corresponding author for this manuscript, should have already received a quote with the article processing fee separately. Please let us know in case this quote has not been received.

Once your article is at Wiley for editorial production, you will receive an email from Wiley's Author Services system, which will ask you to log in and will present you with the publication license form for completion. Within the same system the publication fee can be paid by credit card, an invoice or pro forma can be requested.

Payment of the publication charge and the signed Open Access Agreement form must be received before the article can be published online.

Upon acceptance it is mandatory for you to return the completed payment form. Failure to send back the form may result in a delay in the publication of your paper.

Molecular Systems Biology articles are published under the Creative Commons licence CC BY, which facilitates the sharing of scientific information by reducing legal barriers, while mandating attribution of the source in accordance to standard scholarly practice.

Proofs will be forwarded to you within the next 2-3 weeks.

Thank you very much for submitting your work to Molecular Systems Biology.

Kind regards,

Maria

Maria Polychronidou, PhD
Senior Editor
Molecular Systems Biology